# Defining the RBPome of primary T helper cells to elucidate higher-order Roquin-mediated mRNA regulation

Kai P. Hoefig[1,13], Alexander Reim[2,13], Christian Gallus[3,13], Elaine H. Wong[4], Gesine Behrens[1], Christine Conrad[4], Meng Xu[1], Lisa Kifinger[4], Taku Ito-Kureha[4], Kyra A. Y. Defourny[4,10], Arie Geerlof[5], Josef Mautner[6], Stefanie M. Hauck[7], Dirk Baumjohann[4,11], Regina Feederle[8], Matthias Mann[2], Michael Wierer[2,12 ✉], Elke Glasmacher[3,9 ✉] & Vigo Heissmeyer[1,4 ✉]

Post-transcriptional gene regulation in T cells is dynamic and complex as targeted transcripts respond to various factors. This is evident for the *Icos* mRNA encoding an essential costimulatory receptor that is regulated by several RNA-binding proteins (RBP), including Roquin-1 and Roquin-2. Here, we identify a core RBPome of 798 mouse and 801 human T cell proteins by utilizing global RNA interactome capture (RNA-IC) and orthogonal organic phase separation (OOPS). The RBPome includes Stat1, Stat4 and Vav1 proteins suggesting unexpected functions for these transcription factors and signal transducers. Based on proximity to Roquin-1, we select ~50 RBPs for testing coregulation of Roquin-1/2 targets by induced expression in wild-type or Roquin-1/2-deficient T cells. Besides Roquin-independent contributions from Rbms1 and Cpeb4 we also show Roquin-1/2-dependent and target-specific coregulation of *Icos* by Celf1 and Igf2bp3. Connecting the cellular RBPome in a post-transcriptional context, we find contributions from multiple RBPs to the prototypic regulation of mRNA targets by individual *trans*-acting factors.

[1] Research Unit Molecular Immune Regulation, Helmholtz Center Munich, Munich, Germany. [2] Department of Proteomics and Signal Transduction, Max-Planck-Institute of Biochemistry, Munich, Germany. [3] Institute of Diabetes and Obesity, Helmholtz Center Munich, Munich, Germany. [4] Institute for Immunology, Biomedical Center, Ludwig Maximilians University Munich, Planegg-Martinsried, Germany. [5] Institute of Structural Biology, Helmholtz Center Munich, Neuherberg, Germany. [6] Research Unit Gene Vectors, Helmholtz Center Munich & Children's Hospital, TU Munich, Munich, Germany. [7] Research Unit Protein Science, Helmholtz Center Munich, Munich, Germany. [8] Monoclonal Antibody Core Facility and Research Group, Institute for Diabetes and Obesity, Helmholtz Center Munich, Neuherberg, Germany. [9] Roche Pharma Research and Early Development, Large Molecule Research, Roche Innovation Center Munich, Penzberg, Germany. [10]Present address: Department of Biomolecular Health Sciences, Utrecht University, Utrecht, The Netherlands. [11]Present address: Medical Clinic III for Oncology, Immuno-Oncology and Rheumatology University Hospital Bonn, University of Bonn, Bonn, Germany. [12]Present address: Proteomics Research Infrastructure, University of Copenhagen, Copenhagen, Denmark. [13]These authors contributed equally: Kai P. Hoefig, Alexander Reim, Christian Gallus. ✉email: wierer@biochem.mpg.de; elke.glasmacher@roche.com; vigo.heissmeyer@med.uni-muenchen.de

T lymphocytes as central entities of the adaptive immune system must be able to make critical cell fate decisions fast[1]. To exit quiescence, commit to proliferation, and exert effector functions or form memory they strongly depend on programs of gene regulation[2–6]. Accordingly, they employ extensive post-transcriptional regulation through RBPs or miR-NAs and 3′ end oligo-uridylation or m6A RNA modifications. These RBPs, or RBPs that recognize modifications, directly affect the expression of genes by controlling mRNA stability or translation efficiency[4,7–10]. Previous studies of T helper cells have focused on a small number of RNA-binding proteins, including HuR and TTP/Zfp36l1/Zfp36l2[11–14], Roquin-1/2[15,16] and Regnase-1/4[17,18] as well as some miRNAs like miR-17–92, miR-155, miR-181, miR-125 or miR-146a[19]. Moreover, the first evidence for m6A RNA methylation in this cell type has been provided[9]. Underscoring the relevance for the immune system, loss-of-function of these factors has often been associated with profound alterations in T cell development or functions which caused immune-related diseases[19–22]. Intriguingly, many key factors of the immune system have acquired long 3′-UTRs enabling their regulation by multiple, and often overlapping sets of post-transcriptional regulators[23]. RBPs can also recruit additional co-factors as for example Roquin-1 binds together with Nufip2 to RNA[24], and some of them have antagonistic RBPs like HuR and TTP[25] or Regnase-1 and Arid5a[26]. Such functional or physical interactions together with interdependent binding to the transcriptome create enormous regulatory potential. The major challenge is therefore to integrate our current knowledge about individual RBPs into concepts of higher-order gene regulation that reflect the interplay of different, and ideally of all cellular RBPs.

A prerequisite for studying higher-order post-transcriptional networks is to know the cell type-specific RBPomes that account for differential and dynamic expression of RBPs and mRNP plasticity. To this end several global methods have been developed over the last decade, revealing a growing number of RBPs that may even exceed recent estimates of ~7.5% of the human proteome[27]. RNA interactome capture (RNA-IC)[28] is one widely used, unbiased technique, however, it is constricted by design, intending to identify proteins binding to polyadenylated RNAs. In contrast, orthogonal organic phase separation (OOPS) analyzes all UV-crosslinked protein–RNA adducts from interphases after organic phase separation[29].

The interactions of RBPs with RNA typically involve charge-, sequence- or structure-dependent interactions, and to date over 600 structurally different RNA-binding domains (RBD) have been identified in canonical RBPs of the human proteome[27]. However, global methods also identified hundreds of non-canonical RBPs, which oftentimes contained intrinsically disordered regions (IDRs). Surprisingly, as many as 71 human proteins with well-defined metabolic functions were found to interact with RNA[30] introducing the concept of "moonlighting". Depending on availability from their "day job" in metabolism such proteins also bear the potential to impact RNA regulation. Recent large-scale approaches have increased the number of EuRBPDB-listed human RBPs to currently 2949[31], suggesting that numerous RNA/RBP interactions and cell-type-specific gene regulations have gone unnoticed so far.

In this work, as the first step towards a global understanding of post-transcriptional gene regulation, we experimentally define all proteins that can be crosslinked to RNA in T helper cells. RNA-IC or OOPS identify ~310 or ~1200 proteins in primary CD4+ T cells interacting with polyadenylated transcripts or all RNA species, respectively. Importantly, this dataset now enhances the study of higher-order gene regulation. Testing how the cellular RBPs participate or intervene with post-transcriptional control of

target mRNAs like *Icos* by specific trans-acting factors like Roquin-1/2, we show additional inputs from several other RBPs. These results not only exemplify a previously unrecognized complexity but also imply that post-transcriptional targets integrate simultaneous inputs from all RBPs able to interact with binding sites encoded in their mRNAs.

## Results

**Simultaneous and temporal regulation of Icos through several RBPs.** A prominent example for complex post-transcriptional gene regulation is the inducible T-cell costimulator (Icos), which is essential for humoral immune responses[32–37]. Its mRNA has a long 3′-UTR, which responds in a redundant manner to Roquin-1 and Roquin-2 proteins[16,38–40]. ICOS expression is also repressed by Regnase-1[15,18] and by microRNAs[41,42]. In addition, sites of TTP binding in the ICOS mRNA have been determined by crosslinking and immunoprecipitation[12]. Moreover, the Icos 3′-UTR was proposed to be modified by m6A methylation[43], which could either attract m6A-specific RBPs with YTH domains[44], recruit or repel other RBPs[45], or interfere with base-pairing and secondary structure or miRNA/mRNA-duplex formation[46]. Because of transcriptional and post-transcriptional regulation, Icos expression exhibits a hundred-fold upregulation on the protein level during T cell activation (d1–2), which quickly declines after removal of the TCR stimulus (d3–5) (Fig. 1). To investigate the temporal impact on Icos expression by Roquin-1/2, Regnase-1, m6A, and miRNA regulation we analyzed inducible, CD4-specific inactivation of Roquin-1 together with Roquin-2 (*Rc3h1-2*) or of Regnase-1 (*Zc3h12a*). We also analyzed the inactivation of Wtap, an essential component of the m6A methyltransferase complex[47], or of Dgcr8, which is required for pre-miRNA biogenesis[48]. To this end, we performed tamoxifen gavage on mice expressing a *Cre-ERT2* knockin allele from the CD4 locus[49] together with the floxed, Roquin-1/2 paralogs encoding, *Rc3h1* and *Rc3h2* alleles (Fig. 1a–c) or Regnase-1 encoding *Zc3h12a* (Fig. 1d–f) or *Wtap* (Fig. 1g–i) or *Dgcr8* alleles (Fig. 1j–l). We isolated CD4+ T cells from these mice and expanded them for 5 days. Confirming target deletion on the protein level (Fig. 1c, f, i, l and Supplementary Fig. 1a) we determined a strong negative effect on Icos expression by Roquin-1/2 and Regnase-1 on days 2–5 (Fig. 1a, b and d, e), a moderate positive effect of Wtap on days 2–5 (Fig. 1g, h), and only a small effect of Dgcr8, with an initial tendency of negative (day 1) and later positive effects (days 4–5) (Fig. 1j, k). We next asked whether T cell activation affects the expression levels of known regulators of Icos, as well as other RBPs to establish temporal compartmentalization. To do so, we monitored the expression of a panel of RBPs in mouse CD4+ T cells over the same time course (Fig. 1m–q). Indeed, we revealed or confirmed fast upregulation of RBPs as determined with pan-Roquin, Nufip2[24], Fmrp, Fxr1, Fxr2, TTP/Zfp36[12], pan-Ythdf (Supplementary Fig. 1b), or Celf1-specific antibodies, but also slower accumulation as demonstrated using Regnase-1 or Rbms1-specific antibodies (Fig. 1m, o). There was also downregulation of RBPs as shown with pan-Ago[50] and Cpeb4-specific antibodies (Fig. 1m and p). Of note, we also observed signs of post-translational regulation showing incomplete or full cleavage of Roquin-1/2 or Regnase-1 proteins[15,18], respectively (Fig. 1m), and the induction of a slower migrating band for Celf1, likely phosphorylation[51] (Fig. 1q). Factors with the potential to cooperate, as has been reported for Roquin-1/2 and Nufip2[24] or Roquin-1 and Regnase-1[15], showed overlapping temporal regulation (Fig. 1m), suggesting reinforcing effects for their shared targets, such as Icos.

Together these data indicate that mRNA targets can respond to simultaneous inputs from several RBPs, which are aligned by dynamic expression and post-translational regulation and can

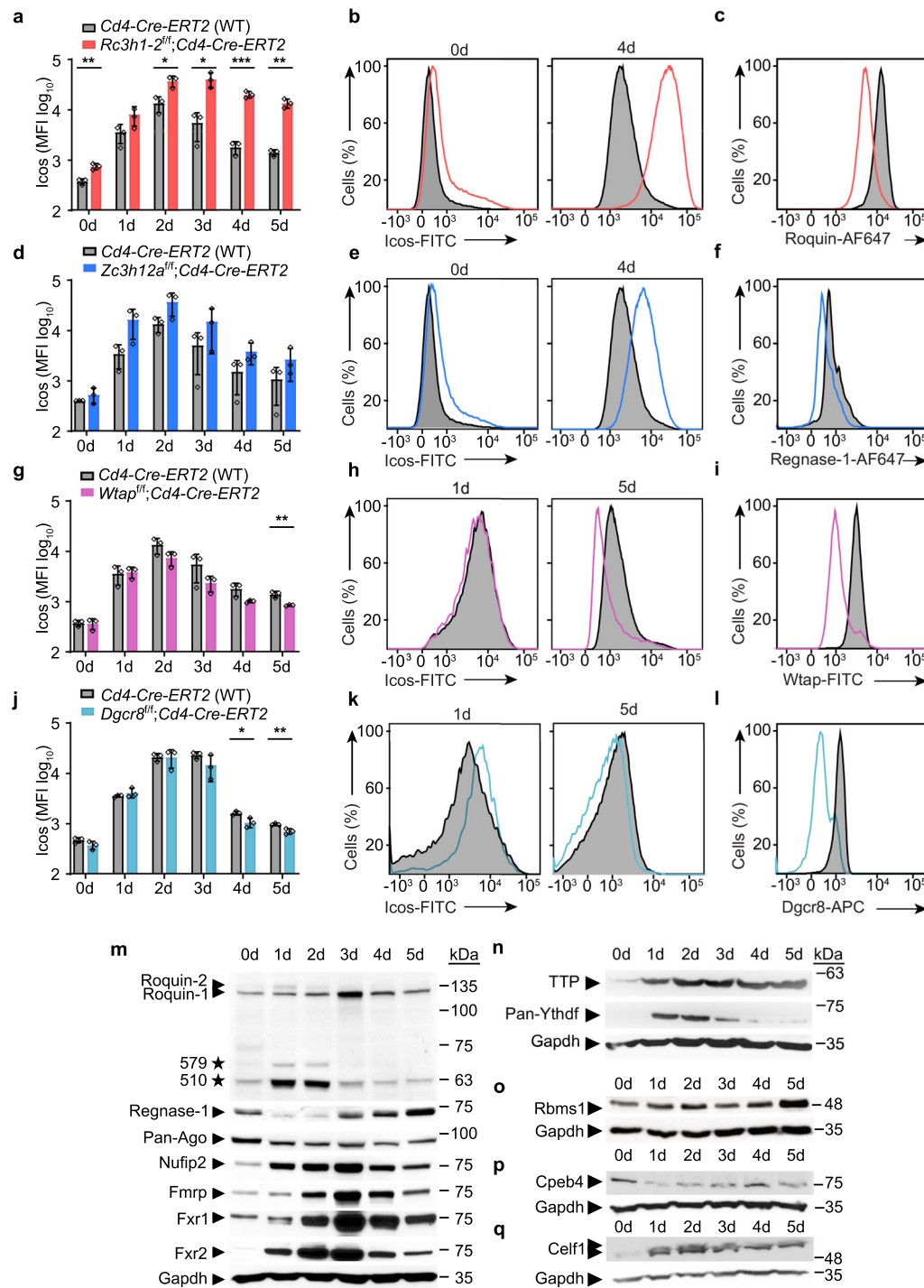

**Fig. 1 Icos responds to simultaneous inputs from several post-transcriptional regulators.** Naive CD4$^+$ T cells (d0) from tamoxifen-gavaged mice of the indicated genotypes were activated by anti-CD3/anti-CD28 (day1–2) and subsequently cultivated in IL-2 containing media (d4–5). **a**, **d**, **g**, **j** Bar diagrams show the average of daily flow cytometric measurements of Icos expression with quantified mean fluorescence intensities (MFI) over a 5-day period to analyze changes due to inducible inactivation of Roquin-1 and Roquin-2 (*Rc3h1* and *Rc3h2*), Regnase-1 (*Zc3h12a*) Wtap or Dgcr8. Significance was calculated using the unpaired *t*-test (two-tailed) for data from three independent experiments using one mouse per genotype (*n* = 3). Error bars, mean ± s.d. **b**, **e**, **h**, **k** Representative histograms of Icos expression at the specified days. **c**, **f**, **i**, **l** Histograms confirming depletion of the respective target proteins. **m–q** Western blots showing patterns of dynamic RBP regulation after anti-CD3/CD28 mediated T cell activation (*n* = 2). significance *$p$ value = 0.01–0.05; **$p$ value = 0.001–0.01; ***$p$ value = <0.001. Calculated $p$ values: (**a**) 0d $p$ = 0.005, 2d $p$ = 0.016, 3d $p$ = 0.012, 4d $p$ = 0.0005, 5d $p$ = 0.0017, (**g**) 5d $p$ = 0.009, (**j**) 4d $p$ = 0.022, 5d $p$ = 0.009. Source data are provided as a Source Data file.

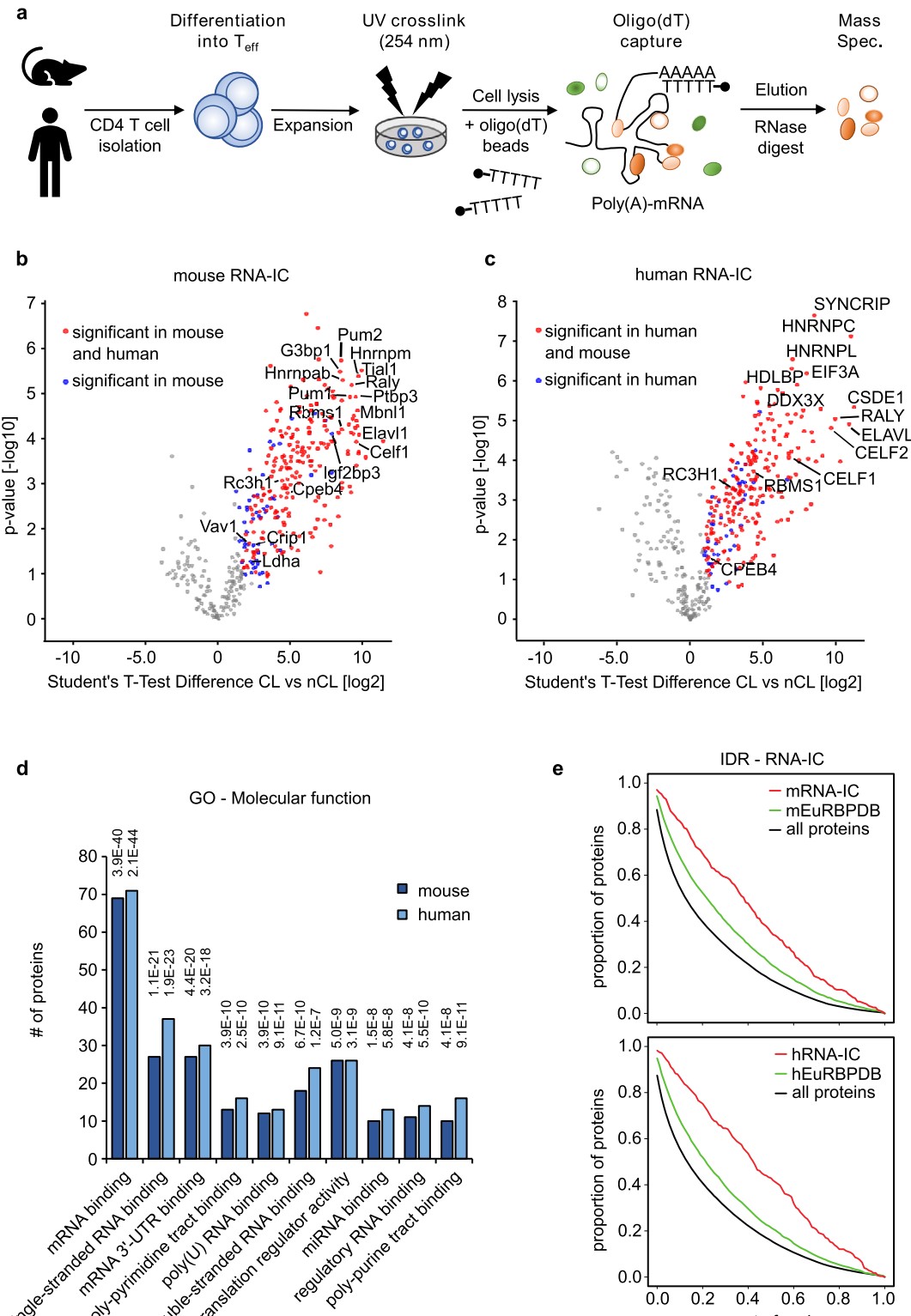

orchestrate redundant, cooperative, and antagonistic effects into a coordinated higher-order regulation.

**The RNA-IC-identified proteins of mouse and human T helper cells.** To analyze the post-transcriptional network, we set out to investigate the RNA-binding protein signature in primary mouse and human CD4$^+$ T cells. To identify mRNA-binding proteins we first performed mRNA capture experiments on $2 \times 10^7$ CD4$^+$ T effector cells (Teff) expanded under $T_H0$ culture conditions, which

omit cytokines and antibodies that skew them into specific subsets (Fig. 2a). Pull-down with oligo-dT beads enabled the enrichment of mRNA-bound proteins, which were increased in response to preceding UV irradiation of the cells, as determined by silver staining (Supplementary Fig. 1c). Reverse transcription-quantitative PCR (RT-qPCR) confirmed the recovery of specific mRNAs, such as for the housekeeping genes *Hprt* and *β-actin*. Both mRNAs were enriched at least 2–3 fold after UV crosslink, but there was no detection of non-polyadenylated 18S rRNA

**Fig. 2 The T helper cell RBPome of polyadenylated RNAs.** $T_H0$ cultures from three mice or three human donors were used as biologic replicates ($n = 3$) to investigate proteins interacting with mRNA. **a** Schematic illustration of the RNA-interactome capture (RNA-IC) method that was carried out to identify RBPs from mouse and human CD4+ T cells. **b**, **c** Volcano plots from two-sided Student's T-test analysis using a permutation-based FDR method for multiple hypothesis corrections showing the −log10 p-value plotted against the log2 fold-change comparing the RNA-capture from crosslinked (CL) mouse CD4+ T cells (**b**) or human CD4+ T cells (**c**) versus the non-crosslinked (nCL) control. Red dots represent proteins significant at a 5% FDR cut-off level in both mouse and human RNA-capture experiments and blue dot proteins were significant only in mice or humans, respectively. **d** Enrichment analysis of GO molecular function terms of significant proteins in mouse or human RNA capture data. The 10 most enriched terms in mouse (dark blue) and the respective terms in human (light blue) are shown. The y-axis represents the number of proteins matching the respective GO term. p-values were calculated using the hypergeometric distribution and were adjusted by Benjamini–Hochberg multiple testing correction. Numbers above each term depict the adjusted p-value. **e** Distribution of IDRs in all Uniprot reviewed protein sequences (black line), in proteins of the mouse EuRBPDB database (green line), and in proteins significant in the mouse RNA-IC experiment (red line). The same plot is shown for human data at the bottom. According to two-sided Kolmogorov–Smirnov testing, the IDR distribution differences between RNA-IC (red lines) and all proteins (black lines) are highly significant in mice and man and reach the smallest possible p-value ($p < 2.2 \times 10^{-16}$).

(Supplementary Fig. 1d). Focusing on protein recovery, we determined greatly enriched polypyrimidine tract-binding protein 1 (Ptbp1) RBP compared to the negative control β-tubulin in mRNA capture experiments using the EL-4 thymoma cell line (Supplementary Fig. 1e). Next, we performed mass spectrometry (MS) on captured proteins from murine and human T cells (Fig. 2b, c). Quantifying proteins bound to mRNA in crosslinked (CL) versus non-crosslinked (nCL) samples we defined a total of 312 mouse (Fig. 2b) and 308 human mRNA-binding proteins (mRBPs) (Fig. 2c) with an overlap of ~70% (Supplementary Data 1), which is in concordance with the overlap of all listed RBPs for these two species in the eukaryotic RBP database (http://EuRBPDB.syshospital.org). Gene ontology (GO) analysis identified the term 'mRNA binding' as most significantly enriched (Fig. 2d). The top 10 GO terms in the mouse were also strongly enriched in the human dataset with comparable numbers of proteins assigned to the individual GO terms in both species (Fig. 2d). RBPs not only bind RNA through classical RBDs, but RNA-interactions can also map to IDRs[52]. Furthermore, low complexity regions (LCRs) have also been reported to be over-represented in RBPs[28]. Indeed, IDRs (Fig. 2e) and LCRs (Supplementary Fig. 1f) were strongly enriched protein characteristics of mouse and human RNA-IC-identified RBPomes. We wondered how much variation existed in the composition of RBPs between different T helper cell subsets. Performing RNA-IC experiments using in vitro generated and phenotypically characterized mouse and human iTreg cells (Supplementary Fig. 2a–d), we find an overlap of 96% or 90% with the respective mouse or human iTreg with Teff RBPomes, suggesting that the same RBPs bind to the transcriptome in different T helper cell subsets (Supplementary Data 1 and Supplementary Fig. 2e). Nevertheless, 47 or 48 proteins were exclusively identified in mouse or human effector T cells, respectively, and 10 or 28 proteins were only found in mouse or human iTreg cells, respectively (Supplementary Fig. 2f). Taken together, these findings suggest that iTreg cells, as an example for T helper cell subset specialization, differentially express eight mouse and 20 human RBPs that do not overlap between the two species (Supplementary Fig. 2f).

**The OOPS-identified proteins of mouse and human T helper cells.** We then attempted to extend and confirm the T cell RBPome with a second approach that utilizes a different biochemical principle to experimentally enrich proteins crosslinked to RNA by employing orthogonal organic phase separation (OOPS)[29]. Similar to interactome capture, OOPS preserves cellular protein/RNA interactions by UV crosslinking of intact cells. The physicochemical properties of the resulting adducts direct them towards the interphase in the organic and aqueous phase partitioning procedure (Fig. 3a). Following several cycles of interphase transfer and phase partitioning, RNase treatment

releases RNA-bound proteins into the organic phase, making them amenable to mass spectrometry[29,53]. Evaluating the method, we selected the UV dose that removes 75% of the total RNA from the aqueous phase[29] (Supplementary Fig. 3a) and investigated selected RNAs and proteins from purified inter-phases derived from CL and nCL MEF cell samples (Supplementary Fig. 3b, c). RNAs with crosslinked proteins purified from interphases hardly migrated into agarose gels, but regained normal migration behavior after protease digest, as judged from the typical 18S and 28S rRNA pattern (Supplementary Fig. 3b). Conversely, known RBPs like Roquin-1 and Gapdh appeared after crosslinking in the interphase and could be recovered after RNase treatment from the organic phase (Supplementary Fig. 3c). Utilizing the same cell numbers and culture conditions of T cells, this method identified in total 1255 and 1159 significantly enriched RBPs for mouse or human T cells, respectively, when comparing CL and nCL samples (Supplementary Data 1). The overlap between both organisms was 55% (Fig. 3b) and 60% (Fig. 3c) in relation to the individual mouse and human RBPomes. Although glycosylated proteins are known to also accumulate in the interphase[29], we experimentally verified that they did not migrate into the organic phase after RNase treatment (Supplementary Fig. 3d, e). Analyzing OOPS-derived RBPomes for gene ontology enrichment using the same approach as for RNA-IC the GO term 'mRNA binding' was again most significantly enriched in mice and humans (Fig. 3d). The top 10 GO terms were RNA related and six of them overlapped with those identified for RNA-IC-derived RBPomes. High similarity between mouse and human RBPomes becomes apparent by the similarity in all GO categories, including 'molecular function' (Fig. 3d), 'biological process', and 'cellular component' (Supplementary Fig. 4). Although our OOPS approach exceeded by far the quantity of RNA-IC identified RBPs, the number of ~1200 RBPs well-matched published RBPomes of HEK293 (1410 RBPs), U2OS (1267 RBPs), and MCF10A (1165 RBPs) cell lines[29].

**Defining the core T helper cell RBPome.** To define a T helper cell RBPome we first made sure that neither RNA-IC nor OOPS preferentially identified high abundance proteins (Fig. 4a, b). In comparison to total proteome measurements, OOPS-identified RBPs spanned the whole range of protein expression without apparent bias. In general, this was also true for RNA-IC, with a tendency to more abundantly expressed proteins. This however might be a true effect since messenger RNA-binding RBPs have been reported to be higher expressed compared to other RBPs[27]. We used the recently established comprehensive eukaryotic RBP database as a reference to compare OOPS- and RNA-IC-identified canonical and non-canonical RBPs from mouse and human CD4+ T cells. The numbers of proteins in the mouse T helper cell RBPomes created by RNA-IC and OOPS ranging from

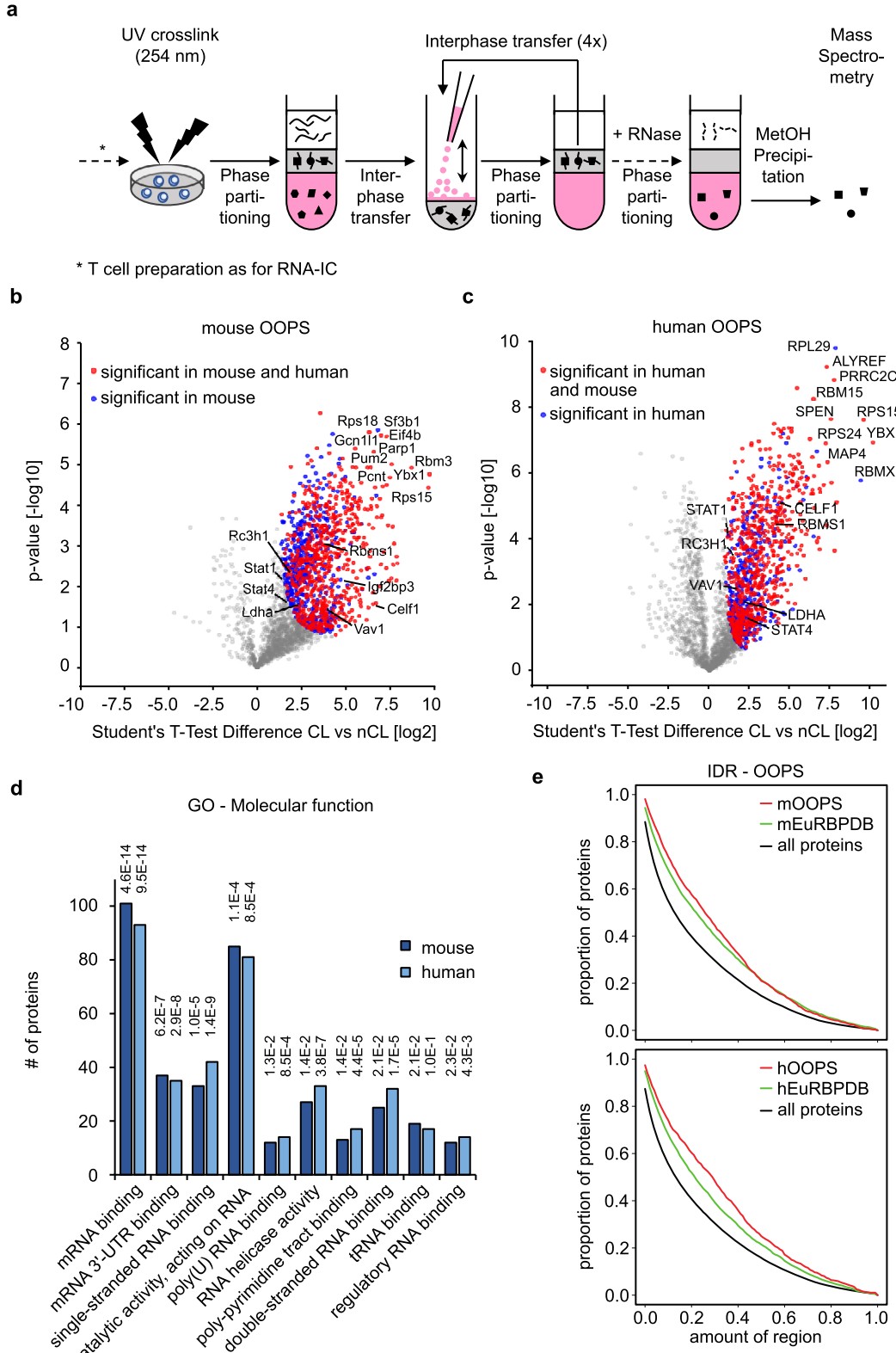

312 to 1255 made up 10–40% of all listed EuRBPDB proteins, respectively (Fig. 4c). OOPS-identified T cell RBPs outnumbered those from RNA-IC experiments by a factor of four, which was predominantly due to the eight times higher number of non-canonical RBPs. Interestingly though, there were also twice as many canonical RBPs significantly enriched by OOPS (Fig. 4c). Analyzing the 10 most abundantly annotated mouse RBDs

(comprising 26–224 RBP family members) showed that RNA-IC and OOPS often identified the same canonical RBPs (Supplementary Tables 1 and 2), however at least equal, higher, or much higher numbers were detected in OOPS samples depending on the specific RBD (Fig. 4d). These findings underscored that the OOPS method is likely more sensitive and by design recovered RBPs from additional, non-polyadenylated RNAs. The data also

**Fig. 3 The global RNA-bound proteome of T helper cells.** $T_H0$ cultures from three mice or four human donors were used as biologic replicates ($n = 3$ or $n = 4$) to characterize proteins interacting with RNA. **a** Schematic overview of the OOPS method[29] with an increased number (5) of phase partitioning cycles. **b** and **c** Volcano plots from two-sided Student's $T$-test analysis using a permutation-based FDR method for multiple hypothesis corrections showing the $-\log_{10}$ $p$-value plotted against the $\log_2$ fold-change comparing the organic phase after RNase treatment of the interphase of OOPS experiments of the crosslinked mouse (**b**) or human CD4$^+$ T cells (**c**) versus the non-crosslinked sample. Red dots represent proteins significant at a 5% FDR cutoff level in both mouse and human OOPS experiments and blue dots represent proteins significant only in mice or humans, respectively. **d** Enrichment analysis of GO molecular function terms of significant proteins in mouse or human OOPS data. Enriched terms are depicted and were calculated as described for RNA-capture data in Fig. 2d. **e** Distribution of intrinsically disordered regions in all Uniprot reviewed protein sequences (black line), in proteins in the mouse EuRBPDB database (green), and in proteins significant in the mouse OOPS data (red line). The same plot is shown for human data at the bottom. According to two-sided Kolmogorov–Smirnov testing, the IDR distribution differences between RNA-IC (red lines) and all proteins (black lines) are highly significant in mice and man and reach the smallest possible $p$-value ($p < 2.2 \times 10^{-16}$).

show that our RNA-IC-derived RBPomes are mostly specific but incomplete. These conclusions are also supported by highly similar results obtained for the human CD4$^+$ T cell RBPome (Fig. 4e, f). In a four-way comparison of mouse and human RBPs identified by OOPS and RNA-IC (Fig. 4g), we conservatively defined all proteins that were identified by at least two datasets as 'core CD4$^+$ T cell RBPomes' discovering 798 mice and 801 human RBPs in this category (Supplementary Data 1). A sizable number of 519 mouse and 424 human proteins were exclusively enriched by the OOPS method, of which more than 50% of the proteins of both subsets matched to mouse or human EuRBPDB-listed annotations (Supplementary Data 1). These findings suggested that genuine RBPs are found even outside of the intersecting set of OOPS and RNA-IC identified proteins and that the definition of RBPomes profits from employing different biochemical approaches. We further compared published human OOPS data sets from the embryonic kidney (HEK293), osteosarcoma (U2OS), and mammary epithelial (MCF10a) cell lines[29] with our dataset from primary human CD4$^+$ T cells (Fig. 4h). The four-way comparison shows that although similar numbers of RBPs were identified overall, the number of uniquely identified RBPs was almost three times higher in CD4$^+$ T cells than in each of the cell lines (Fig. 4h, left panel). Of the 439 CD4$^+$ T cell unique RBPs 294 were newly discovered and 145 were previously annotated (Fig. 4h, right panel and Supplementary Data 1). The annotated RBPs can be further divided into 92 canonical and 53 non-canonical RBPs (Supplementary Data 1). We interpret the result such that RBPomes are strongly affected by tissue-specific expression of RBPs and/or RBP activity in the presence or absence of post-translational modifications, substrates, and co-factors.

**T cell signaling proteins with unexpected RNA-binding function**. Some of the identified RBPs of the core proteome including Stat1, Stat4, and Crip1 are not expected to be associated with mRNA in cells. We, therefore, established assays to confirm the RNA-binding of these candidates. To do so, GFP-tagged candidate proteins were overexpressed in HEK293T cells, which were UV cross-linked, and extracts were used for immunoprecipitations with GFP-specific antibodies. Using SDS–PAGE and protein blotting and detection with either anti-GFP antibodies or oligo(dT) probes verified the pull-down of GFP-tagged proteins (Fig. 5a, left panel) and the association with mRNA (Fig. 5a, right panel) for the RBPs Roquin-1 and Rbms1 as well as for the lactate dehydrogenase (Ldha) protein, a metabolic enzyme with known ability to also bind RNA[28] (Fig. 5a). Via this approach, the determined RNA association of Stat1, Stat4, and Crip1 was indeed confirmed (Fig. 5b). It appeared less pronounced as compared to prototypic RPBs but was similar with regard to Ldha (Fig. 5a, b). We then tested human and mouse STAT1 and STAT4 proteins as purified recombinant proteins in RNA-EMSAs with in vitro transcribed TSU lncRNA. This lncRNA is

expressed in human cells and early work revealed a sequence-specific recognition by STAT1 using extracts of STAT1-transfected cell extracts[54,55]. Performing RNA-EMSA without and with competitor RNA we showed binary interaction of mouse and human STAT1 and STAT4 that was at least partially resistant to unspecific competition (Fig. 5c). Our results thereby excluded a requirement for additional factors or signal-induced STAT1 or STAT4 protein modification in eukaryotic cells or any indirect contribution from cell extracts in these RBP/RNA interactions. These findings support a potential moonlighting function of these signaling proteins. To address a regulatory function for STAT1 or STAT4 protein binding to RNA, we established a dual luciferase assay to investigate the impact of the different proteins on the expression of the renilla luciferase when they were tethered to its mRNA via an artificial 3′-UTR (Fig. 5d). We utilized the λN/5xboxB system[56] and confirmed the expression of fusion proteins with a newly established λN-specific antibody (Fig. 5e, f). Importantly, Stat1 and Stat4 repressed luciferase function almost to the same extent as the known negative regulators Pat1b and Roquin-1, or other known RBPs, such as Celf1, Rbms1, and Cpeb4 (Fig. 5g), and this repression reduced the abundance of the boxB containing renilla luciferase mRNA (Fig. 5h). λN-Crip1 and λN-Vav1 expression did neither exert a positive nor a negative effect, since their relative luciferase expression appeared unchanged compared to cells transfected to express only the λN polypeptide (Fig. 5g). These data suggest that the transcription factors Stat1 and Stat4, which we defined here as part of the T helper cell RBPome, not only have the capacity to bind mRNA but can also exert RNA regulatory functions. While Vav1 was identified in mouse and human T cells by OOPS and RNA-IC, the tethering assay did not reveal obvious regulation, indicating a more specialized function of this new RBP.

**Analyzing higher-order post-transcriptional regulation**. Since large-scale parallel screens with primary T cells are highly challenging, we devised an experimental strategy to reduce the RBPome to RBPs that are likely to antagonize or cooperate with the Roquin-1 RBP in the repression of its target mRNAs. To this end, we performed 'BioID' experiments to define the cellular proteins that are physically close to Roquin-1 (Fig. 6a). In this proximity-based labeling method, we expressed a Roquin-1 BirA* fusion protein to identify proteins that reside within a short distance of ~10 nm[57] in T cells (Fig. 6a). In this dataset we sought for matches with the T cell RBPome (Fig. 4g) to identify proteins that shared the features, 'RNA-binding' and 'Roquin-1 proximity'. We first verified that the mutated version of the biotin ligase derived from *E. coli* (BirA*) which was N-terminally fused to Roquin-1 or GFP was able to biotinylate lysine residues in Roquin-1 or other cellular proteins (Fig. 6b) but did not interfere with the ability of Roquin-1 to downregulate Icos (Fig. 6c). Doxycycline-induced BirA*-Roquin-1 compared to BirA*-GFP expression in CD4$^+$ T cells significantly enriched biotin labeling

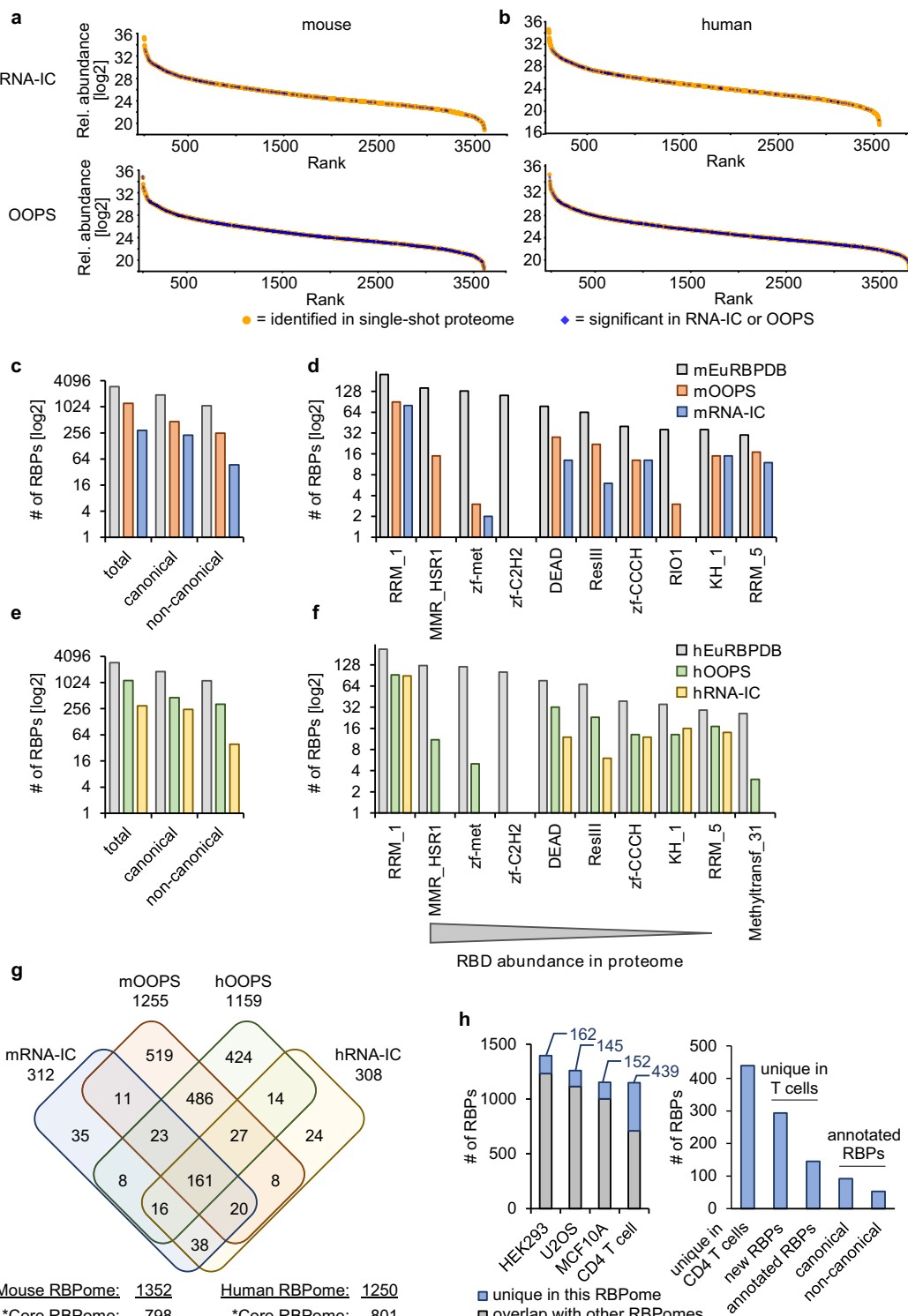

of 64 proteins (Supplementary Data 2), including Roquin-1 (Rc3h1) itself or Roquin-2 (Rc3h2) (Fig. 6d) as well as previously identified Roquin-1 interactors and downstream effectors, such as Ddx6 and Edc4[38], components of the Ccr4/Not complex[58,59] and Nufip2[24] (Fig. 6d). More than half of all proteins in proximity to Roquin-1 were also part of the defined RBPome (Fig. 6e and Supplementary Table 3). Roquin-1 may come close to other RBPs while being bound to RNA, however, proximity labeling will take place in all different parts of the cell independent of RNA

binding. Increasing the BioID list above with additional proteins that we found in proximity to Roquin-1 when establishing and validating the BioID method in fibroblasts (Supplementary Fig. 5a–d), we arrived at 143 proteins (Supplementary Data 2) of which 96 (67%) were part of the RBPome (Supplementary Figs. 5e, 6a, and Supplementary Table 4). From these, we cloned 46 candidate genes of interest (GOI) in the context of N-terminal GFP fusions and determined GFP expression in HEK293T cells (Supplementary Fig. 6b). We then transduced CD4+ T cells and

**Fig. 4 Defining the mouse and human CD4$^+$ T cell RBPomes.** The RBPomes of Figs. 2 and 3 were inspected for enrichment of abundant proteins, representation of specific RBD, overlapping, and uniquely identified proteins. **a** Relative abundance (log$_2$) of proteins identified in a single-shot mouse proteome (orange dots) plotted by their rank from highest to lowest abundant protein. RNA-binding proteins detected by RNA capture (top plots) or by OOPS (bottom plots) are highlighted as blue diamonds. **b** Same plots as shown in (**a**) for human data. **c**, **e** The recently established database EuRBPDB was used as a reference for eukaryotic RBPs to establish the numbers of canonical and non-canonical RBPs identified by RNA-IC or OOPS on the mouse (**c**) or human (**e**) CD4$^+$ T cells. **d**, **f** The occurrence of RBDs in RNA-IC and OOPS-identified RBPs were analyzed in comparison with the 10 most abundant motifs described for the mouse (**d**) or human (**f**) proteome. **g** Venn diagram using four datasets for RBPs in CD4$^+$ T cells as determined by RNA-IC and OOPS in mouse and human cells. We defined the core RBPomes to contain proteins present in at least two datasets. **h** Result from a four-way Venn diagram comparing published OOPS-identified RBPome data sets[29] from the embryonic kidney (HEK 293), mammary epithelial (MCF10A), and bone osteosarcoma epithelial (U2OS) cell lines with the OOPS-derived RBPome from primary human CD4$^+$ T cells (left panel). The right panel depicts results obtained by matching 439 T cell-unique identified RBPs against the human EuRBPDB.

analyzed the effects of GFP-GOI expression on endogenous Roquin-1 targets (Supplementary Fig. 7a). CD4$^+$ T cells were used from mice with *Rc3h1*$^{fl/fl}$;*Rc3h2*$^{fl/fl}$;rtTA alleles in combination with (iDKO) or without the *Cd4-Cre-ERT2* allele (WT) allowing induced inactivation of Roquin-1 and -2 by 4′-OH-tamoxifen treatment. The Roquin-1 targets Icos, Ox40, Ctla4, IκB$_{NS}$, and Regnase-1 became strongly derepressed in induced double-knockout (iDKO) T cells (Supplementary Fig. 7b). This elevated expression was corrected to wild-type levels in iDKO T cells that were retrovirally transduced and doxycycline-treated to express GFP-Roquin-1 (Supplementary Fig. 7b). The target expression in WT T cells was only moderately reduced through ectopic expression of GFP-Roquin-1 (Supplementary Fig. 7b). For the majority of the 46 candidate genes, induced expression in WT or iDKO CD4$^+$ T cells did not alter the expression of the five analyzed Roquin-1 targets, exemplified here by the results obtained for Vav1 (Supplementary Fig. 7c, d). Interestingly, we identified a new function for Rbms1 (transcript variant 2), specifically upregulating Ctla4 (Supplementary Fig. 7e, f). Furthermore, we demonstrated that Cpeb4 strongly upregulates Ox40 and, most strikingly, in the same cells Cpeb4 repressed Ctla4 levels (Supplementary Fig. 7g, h). While these findings are noteworthy, they occurred in a Roquin-1-independent manner. In contrast to these effects, we discovered a higher-order regulation of Icos by Igf2bp3 (Fig. 7a, b). Interestingly, Igf2bp3, an unconventional reader of RNA methylation[60], was consistently identified in mice but not in human CD4$^+$ T cell RBPomes by RNA-IC and OOPS and hence may have divergent functions in these two species. An even stronger Roquin-1 dependent increase of Icos occurred upon induced expression of Celf1 (Fig. 7c, d). In wild-type T cells Celf1 clearly upregulated Icos, Ctla4, and Ox40 expression but not the *Nfkbid* mRNA encoded IkBNS protein expression and this function was obliterated in Roquin-1-deficient iDKO cells (Fig. 7c, d). While Igf2bp3 and Roquin-1 shared a strictly cytoplasmic localization and enrichment in BFP-Ddx6-labeled P-bodies (Supplementary Fig. 8a), the majority of GFP-Celf1 was nuclear. Only a small fraction of the protein was cytoplasmic, where it colocalized with Roquin-1 and Ddx6 in P-bodies (Fig. 7e). The antagonistic effect could not be explained by Celf1-mediated repression of Roquin-1 on the protein or mRNA level (Supplementary Fig. 8b, c). Vice versa, Roquin-1 KO also did not affect Celf1 mRNA levels (Supplementary Fig. 8d). Instead, the observed antagonistic effect likely involved simultaneous or mutually exclusive binding of Celf1 and Roquin-1 to the same mRNAs, since we determined strong interaction of Celf1 with *Icos* and *Ctla4* mRNAs in RNA-IP experiments, but not with the *Nfkbid* mRNA (Fig. 7f). In conclusion, the combination of protein-centric and RNA-centric global approaches enabled us to discover higher-order functional interactions as shown for the Roquin-1/2-dependent regulation of the costimulatory receptor Icos by Igf2bp3 or Celf1.

## Discussion

The work on post-transcriptional gene regulation in T helper cells has focused on some miRNAs and several RNA-binding proteins, and few reports described m6A RNA methylation in this cell type. Although arriving at a more or less detailed understanding of individual molecular relationships and regulatory circuits, this isolated knowledge assembles into a very incomplete picture. Defining the human and mouse T helper cell RBPomes has now opened the stage, allowing to work towards understanding connections, deciphering complexity and principles of post-transcriptional regulatory networks in these cells.

RNA-IC and OOPS are two complementary methods to define RBPs on a global scale. While RNA-IC mostly queries for proteins bound to polyadenylated RNAs, OOPS captures the RNA-bound proteome in its whole. Applying both methods to T helper cells of two different organisms allowed us to cross-validate the results from both methods and solidified our description of the core mouse and human T helper cell RBPome. While the vast majority of RNA-IC-identified CD4$^+$ T cell RBPs were previously known RNA binders, OOPS typically confirmed and profoundly expanded these results (Supplementary Tables 1 and 2), and more than half of OOPS identified proteins that were exclusively found in mouse or man were EuRBPDB-listed.

Strikingly, the signaling proteins Stat1 and Stat4 were identified by mouse RNA-IC and human OOPS and were just below the cut-off (0.05 FDR, >2fold enrichment) in the mouse OOPS dataset, and we could support their RBP function by additional RNA-binding assays. Undoubtedly, the defined human and mouse T cell RBPomes contain many more unusual RBPs that would warrant further investigations. We assume that even RNA interactions of proteins without prototypic RBDs, like the Vav1 and Stat proteins, will have consequences for both binding partners. As the identity of the interacting mRNA(s) is currently unknown, we could only speculate about the post-transcriptional impact. Nevertheless, Stat1 and Stat4, but not Vav1, showed regulatory capacity in our tethering assays. Intriguingly, RNA-binding may impact the function of Stat proteins as transcription factors. Supporting this notion, early results found Stat1 bound to the non-coding, polyadenylated RNA 'TSU', derived from a trophoblast cDNA library, and translocation of Stat1 into the nucleus was reduced after TSU RNA microinjection into HeLa cells[54,55]. In line with this, Vav1 signal-transduction could be altered in an unknown way through its engagement with RNA.

Many 3′-UTRs, which effectively instruct post-transcriptional control, exhibit little sequence conservation between species, and the exact modules which specify regulation are not known. This is for example true for the *Icos* mRNA[24,38]. On the side of the *trans*-acting factors, we find a high similarity between the RBPomes of T helper cells of mouse and human origin, actually reflecting the general overlap of so far determined RBPomes from many cell lines of these species.

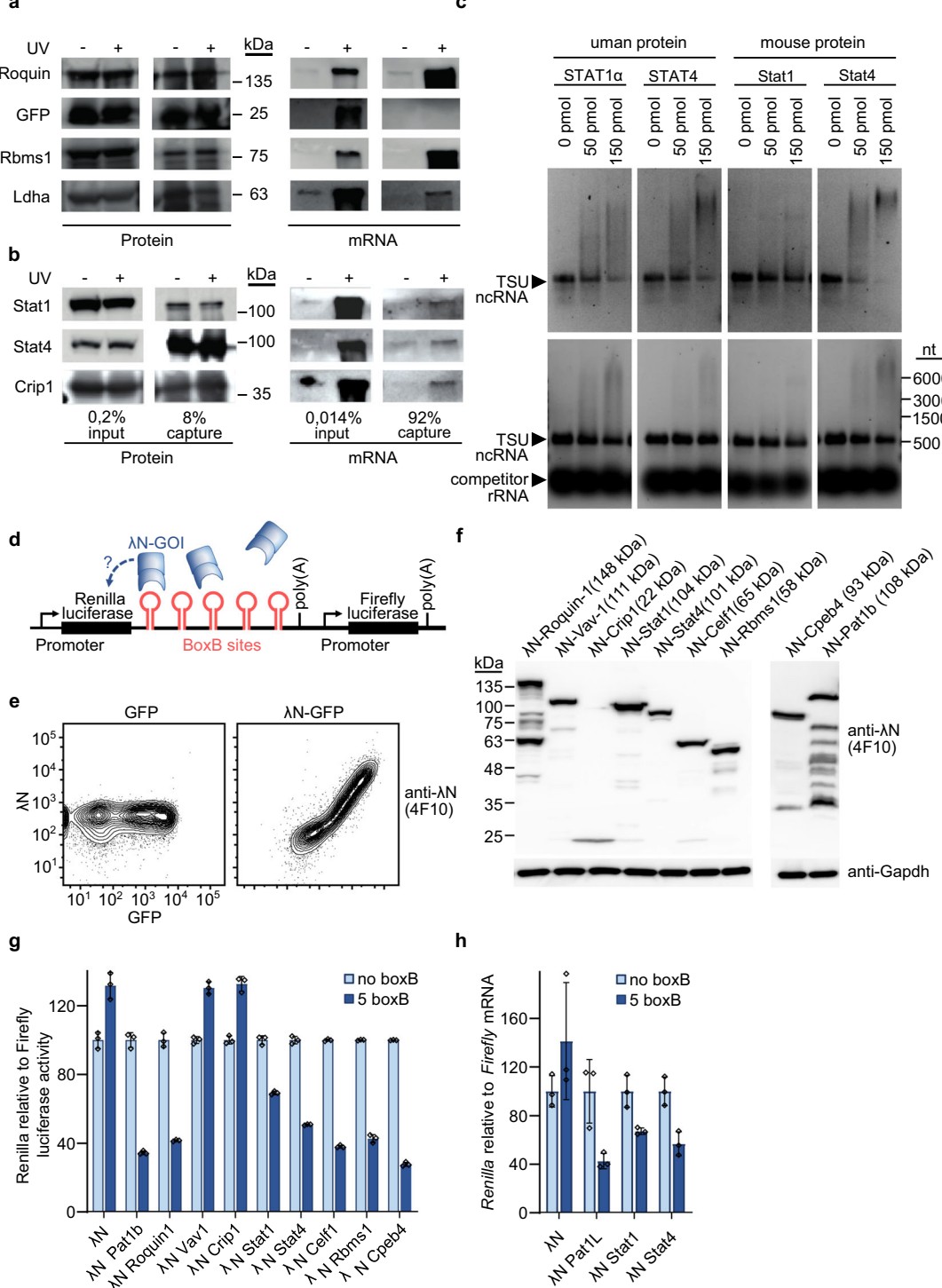

We define the first RBPomes of human and mouse T helper cells and explore avenues of how to make use of this information. Screening a set of candidates from the T cell RBPome for effects on Roquin-1/2 targets, our findings support a concept in which mRNA targets of one *trans*-acting factor are further separated into different "RNA operons" i.e. subsets of RNAs responding to the same post-transcriptional inputs[61]. Such "RNAs operons" can then form "RNA regulons" comprising coordinately regulated mRNA subsets that are aligned to function in the same biological process. Therefore, complex and differential binding of targets by RBPs of the cellular RBPome specify the possible operons and regulons and enable differential functions of the cell. Roquin-1/2

cooperated, coregulated, or antagonized in the regulation of Icos with Regnase-1 and m6A or miRNA functions. Our screening approach added Rbms1 or Cpeb4 as Roquin-1/2-independent regulators of Ctla-4 and/or Ox40. It also revealed that different targets of Roquin-1/2 responded very differently to the expression of specific coregulators, as for example Ctla4 and Ox40 were inhibited or induced by the same RBP, Cpeb4, respectively. Moreover, Celf1 and Igf2bp3 were identified as Roquin-1/2-dependent coregulators in the *Icos* containing RNA operons. Intriguingly, Celf1 only interacted with those target mRNAs of Roquin-1/2 for which it antagonized repression. This suggests that Celf1 may either prevent binding of Roquin-1/2 to some

**Fig. 5 Stat1 and Stat4 bind to RNA and have regulatory potential.** Transfected or recombinant Stat1 and Stat4 were analyzed for interaction with RNA and regulatory potential. **a**, **b** Semi-quantitative identification method for RBPs as described in[75]. In short, GFP-fused proteins were transfected into 293T cells, crosslinked or left untreated, and subsequently immunoprecipitated using an anti-GFP antibody. The obtained samples were divided for protein and RNA detection after transfer to membranes ($n = 2$). **c** Electrophoretic mobility shift assays using purified human STAT1α and STAT4 as well as mouse Stat1 and Stat4 proteins to bind in vitro-transcribed non-coding RNA 'TSU' (481 nt) in the absence (upper panels) and presence (lower panels) of competitor RNA (baker's yeast rRNA) ($n = 2$). **d** Schematic representation of the tethering assay that was used to investigate a possible influence of the genes of interest (GOI) on renilla luciferase expression. The affinity of the λN peptide targets the respective fusion protein to boxB stem-loop structures (5x) in the 3′-UTR of a Renilla luciferase gene where it can exert its function, if exciting. **e** FACS plots using the new monoclonal antibody 4F10 to demonstrate specific λN detection of a λN-GFP fusion protein expressed in 293T cells. **f** Western blot showing the expression of the indicated λN-GOI proteins in 293T cells after transfection ($n = 2$). **g** Tethering assay results performed in HeLa cells as described in (**d**). Two negative controls were implemented, using constructs without boxB sites or λN expression without fusion to a GOI. Each measurement was performed in triplicate and was independently repeated at least twice ($n = 2$). Error bars of three technical replicates, mean ± s.d. **h** Experiment performed as in (**g**), however, Renilla and Firefly RNA levels were assessed by qPCR ($n = 3$). Error bars of biological replicates, mean ± s.d. Source data are provided as a Source Data file.

binding sites or Celf1 may be part of mRNPs with Roquin-1/2 proteins in which it counteracts repression. Together these data indicate an unexpected wealth of possible inputs originating from variable formation of mRNPs on post-transcriptional target mRNAs. These combinatorial compositions reflect the T helper cell RBPome and respond to expression and activity changes of individual components to enable higher-order post-transcriptional gene regulation.

To solve the seemingly simple question of which RBPs regulate which mRNAs in T helper cells, we will require further knowledge about individual contributions, binding sites, and composite *cis*-elements, temporal and interdependent occupancies, interactions among RBPs and with downstream effector molecules. In this endeavor global protein and RNA-centric approaches make fundamental contributions.

## Methods

**Isolation, in vitro cultivation, and transduction of mouse primary CD4+ T cells.** For in vitro cultivation of primary murine CD4+ T cells, mice were sacrificed and spleen, as well as cervical, axillary, brachial, inguinal, and mesenteric lymph nodes, were dissected and pooled. Single-cell suspensions were generated from lymphoid organs and passed through a 100 µm filter under rinsing with T cell isolation buffer (PBS supplemented with 2% FCS and 1 mM EDTA). Erythrocytes were eliminated by incubating cells with TAC–lysis buffer (13 mM Tris, 140 mM NH4Cl, pH 7.2) for 5 min at room temperature. CD4+ T cells were isolated by negative selection using EasySep Mouse CD4+ T cell isolation Kit (19852A, Stem Cell) according to the manufacturer's protocol. CD4+ T cells were cultured in DMEM (41966-029, Invitrogen) T cell culture medium supplemented with 10% FCS (AC-SM-0184, Anprotec), 1% Pen-Strep (15140-122, 10,000 U/ml Penicillin, 10,000 U/ml Streptomycin, Thermo Fisher), 10 mM HEPES-buffer (15630-056, Invitrogen), 1% non-essential amino acids (13-114E, 100× NEAA mixture; Invitrogen), and 50 µM β-mercaptoethanol (31350-010, Invitrogen) without antibodies or cytokines that skew their differentiation into specific T helper cell subsets (TH0 conditions). For activation and differentiation under TH1 conditions the T cells were stimulated with α-CD3 (0.5 µg/ml; cl. 145-2C11, in house production), α-CD28 (2.5 µg/ml; cl. 37.5N, in house production), 10 µg/ml α-IL-4 (cl. 11B11, in house production), and 10 ng/ml IL-12 (554592, BD Pharmingen) and seeded T cells on goat α-hamster IgG (0.05 mg/ml in PBS, 56984, MP Biochemicals) pre-coated six-well (5 Mio cells/ml for 40 h for transductions with retroviruses) or 12-well (1.5 Mio cells/ml for 48 h for expression analyses) plates. The cells were then resuspended and cultured in a medium supplemented with 200 IE/ml recombinant hIL-2 (Proleukin S, Novartis) in a 10% CO2 incubator and expanded for 2–4 days, as indicated. Subsequently, cells were fed with fresh IL-2-containing medium every 24 h and cultured at a density of 0.5–1 × 10^6 cells/ml. For in vitro deletion of floxed alleles of Rc3h1^fl/fl;Rc3h2^fl/fl;Cd4-Cre-ERT2;rtTA3 (but with no effect on the Cre-deficient WT control Rc3h1^fl/fl;Rc3h2^fl/fl;rtTA3) CD4+ T cells were treated with 1 µM 4′OH-tamoxifen (Sigma) for 24 h prior to activation at a cell density of 0.5–1 × 10^6 cells/ml. We performed retroviral transduction 40 h after the start of anti-CD3 (0.25 µg/ml)/CD28 (2.5 µg/ml) activation, T cells were transduced with retroviral particles using spin-inoculation (1 h, 18 °C, 850×g), and after 4–6 h co-incubation of T cells and virus, virus particles were removed and T cells resuspended in T cell medium supplemented with IL-2 as described before. To induce expression of pRetro-Xtight-GFP constructs in rtTA-expressing T cells, the transduced cells were cultured for 16 h in the presence of doxycycline (1 µg/ml) prior to flow cytometry analysis of expression of targets in transduced GFP+ cells.

**In vivo deletion of loxP-flanked alleles and in vitro culture of CD4+ T cells.** For in vitro culture analysis, deletion of Roquin-1/2 (Rc3h1^fl/fl;Rc3h2^fl/fl, Cd4-Cre-ERT2)[62], Regnase-1 (Zc3h12a^fl/fl; Cd4-Cre-ERT2)[63], Wtap (Wtap^fl/fl; Cd4-Cre-ERT2)[47], and Dgcr8 (Dgcr8^fl/fl; Cd4-Cre-ERT2)[48] encoding alleles in Cd4-Cre-ERT2 mice was induced in vivo by oral transfer of 5 mg tamoxifen in corn oil. Two doses of tamoxifen each day were given on 2 consecutive days (total of 20 mg tamoxifen per mouse). Mice with the genotype Cd4-Cre-ERT2 (without floxed alleles) were used for wild-type controls. Mice were sacrificed 3 days after the last gavage and total CD4+ T cells were isolated using the EasySep Mouse T Cell Isolation Kit (19852A, Stem Cell) and activated under TH1 conditions as described above. Animal breeding and experimentation followed the legal approval of the Government of Upper Bavaria (Regierung von Oberbayern, reference numbers 55.2-2532-Vet_02-19-122 and 55.2.2532.Vet_02-19-68). The work was compliant with the relevant ethical regulations for animal testing and research. All animals were housed in a pathogen-free barrier facility in accordance with the Helmholtz Center Munich, the Ludwig Maximilians University Munich institutional, state, and federal guidelines.

**Flow cytometry.** Following in vivo deletion and CD4+ T cell isolation (above) cells were activated and cultured under TH1 conditions. Cells were obtained daily for FACS analysis. The single-cell suspensions were stained with fixable violet dead cell stain (L34955, Thermo Fisher) for 20 min at 4 °C. For the detection of surface proteins, cells were stained with the appropriate antibodies in FACS buffer for 20 min at 4 °C. After staining, cells were acquired on a FACS Canto II (3-laser). The data were further processed with the software FlowJo 10 software (BD Bioscience). The following antibodies were used at a dilution of 1:200: anti-CD4 (cl. GK1.5), anti-CD44 (cl. IM7), anti-CD62L (cl. MEL-14), anti-Icos (cl. C398.4A), anti-Ox40 (cl. OX-86), all from eBioscience, anti-CD25 (cl. PC61, Biolegend).

Effects of doxycycline-induced expression of 46 GFP-GOI fusion proteins in 2 × 10^6 wild-type and Roquin-1/2 iDKO CD4+ T cells were analyzed on day 6 after isolation (compare Supplementary Fig. 7b). First, proteins were treated with a fixable blue dead cell stain (L23105, Invitrogen) and after washing, stained in three panels to interrogate the surface expression of Icos and Ox40 (Icos-PE, clone 7E.17G9/Ox40-APC, clone OX-86; both eBioscience) and to intracellularly measure Ctla4, IκBNS (Ctla4-PE, UC10-4B9; eBioscience/cl. 4C1 rat monoclonal; in house production) as well as Regnase-1 (cl. 15D11 rat monoclonal; in house production). For intracellular staining, cells were fixed in 2% formaldehyde for 15 min at RT, permeabilized by washing in Saponin buffer, and stained with the appropriate antibodies for 1 h at 4 °C. After washing, an anti-rat-AF647 antibody (cl. poly4054; Biolegend) was added for 30 min. All in-house monoclonal antibody supernatants generated at the Helmholtz Center were used at a dilution of 1:10 and all commercial antibodies were diluted 1:200. After additional rounds of Saponin- and FACS buffer washing, the acquisition was performed using an LSR Fortessa device.

**Isolation and differentiation of mouse effector and regulatory T cells for RNA-IC.** Naïve CD4+ T cells were isolated by using Dyna- and Detachabeads (11445D and 12406D, Invitrogen) from spleens and mesenteric lymph nodes of 8–12 weeks old C57BL/6J mice. For iTreg culture, cells were additionally selected for CD62L+ with anti-CD62L (clone: Mel14)-coated beads. All cells were then activated with plate-bound anti-CD3 (using first anti-hamster, 0.05 mg/ml in PBS, 55397, Novartis, then anti-CD3 in solution: clone: 2C11H: 0.1 µg/ml) and soluble anti-CD28 (clone: 37N: 1 µg/ml) and cultured in RPMI medium (supplemented with 10% (vol/vol) FCS, β-mercaptoethanol (0.05 mM, Gibco), penicillin–streptomycin (1 U/ml, Gibco), sodium pyruvate (1 mM, Lonza), 1% non-essential amino acids (11350912, 100× NEAA solution, Gibco), 1% MEM vitamin solution (11120052, 100× solution, Gibco), 1% glutamax (13462629, 100× solution, Gibco), and HEPES pH 7.2 (10 mM, Gibco)). For iTreg cell differentiation we additionally added the following cytokines and blocking antibodies: rmIL-2 and rmTGF-β (R&D Systems, 5 ng/ml), anti-IL-4 (clone 11B11, 10 µg/ml), and anti-IFNγ (clone Xmg-121, 10 µg/ml). All in-house

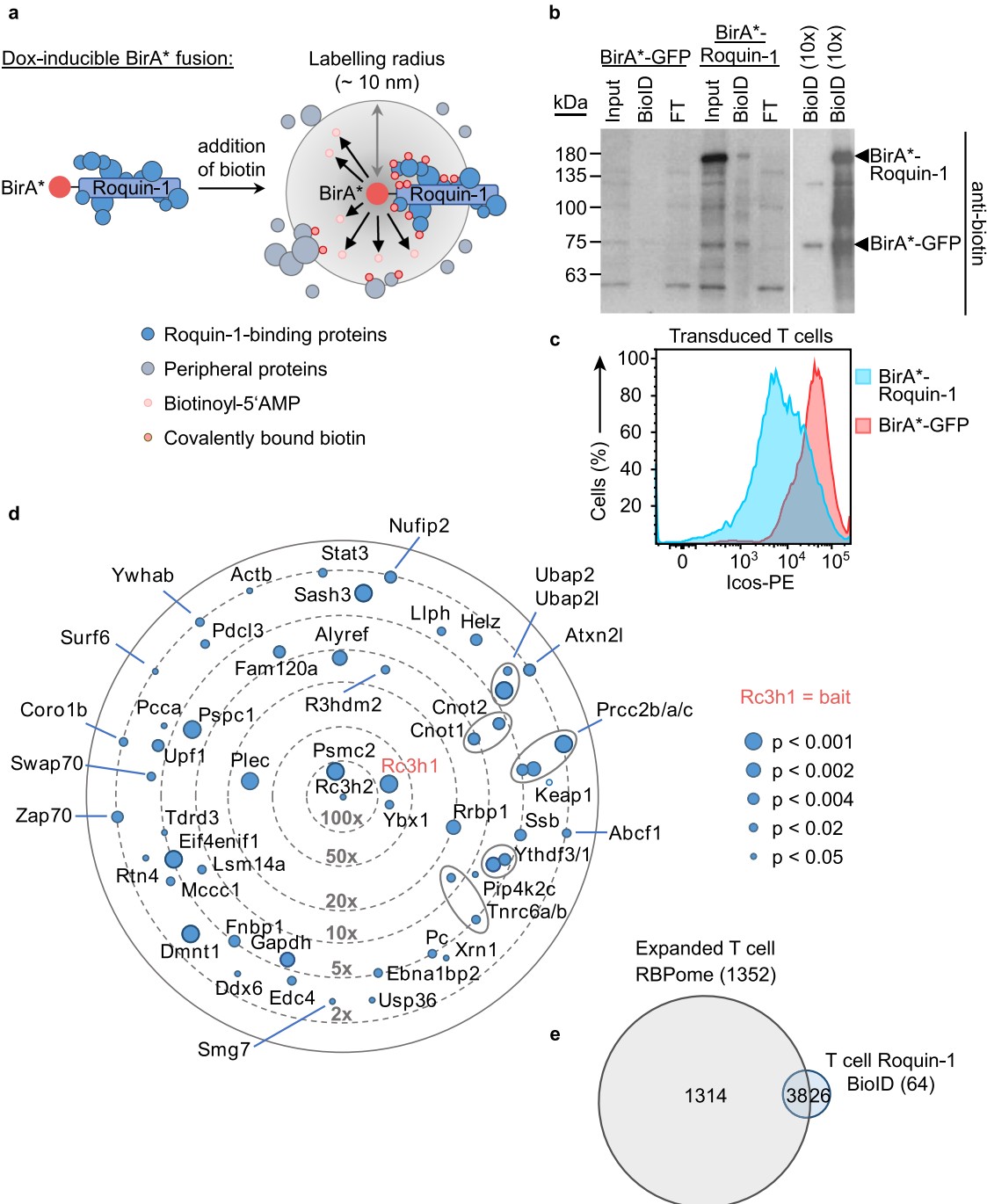

**Fig. 6 Identification of proteins in proximity to Roquin-1 in CD4$^+$ T cells.** Expression of BirA*-Roquin-1 and biotin tagging define a subset of the T cell RBPome in close proximity to Roquin-1. **a** Schematic overview of the BioID method showing how the addition of biotin to the medium leads to the activation of biotin, diffusion of biotinoyl-5'AMP and the biotinylation of the bait (Roquin-1) and all preys in the circumference. **b** Equimolar amounts of protein were loaded onto a PAGE gel for Western blotting applying an anti-biotin antibody. Efficient biotinylation of both baits BirA*-Roquin-1 and BirA*-GFP (control) could be demonstrated ($n = 2$). **c** Histogram showing that transduction of CD4$^+$ T cells with retrovirus to inducibly express BirA*-Roquin-1 lead to the efficient downregulation of endogenous Icos. **d** Identified preys from Roquin-1 BioIDs ($n = 5$) in CD4$^+$ T cells. Depicted are all significantly enriched proteins with the exception of highly abundant ribosomal and histone proteins. A two-sided heteroscedastic Student's *T*-test analysis was performed. Dot sizes equal *p*-values and positioning towards the center implies increased x-fold enrichment over BirA*-GFP BioID results. **e** Venn diagram showing the overlap of RBPs from the CD4$^+$ T cell RBPome with the proteins identified by Roquin-1 BioID in T cells. Source data are provided as a Source Data file.

produced antibodies were obtained in collaboration with and from Regina Feederle (Helmholtz Center Munich). After differentiation for 36–48 h cells were expanded for 2–3 days. iTreg cells were cultured in RPMI and 2000 units Proleukin S (02238131, MP Biomedicals) and T$_{eff}$ cells with 200 units Proleukin S. We only used iTreg cells for experiments if samples achieved at least 80% Foxp3 positive cells (00552300, Foxp3 Staining Kit, BD Bioscience). EL-4 T cells were cultured in the same medium as primary T cells. HEK293T cells were cultured in DMEM

(supplemented with 10% (vol/vol) FCS, penicillin–streptomycin (100 U/ml, Gibco) and Hepes, pH 7.2 (10 mM, Gibco)).

**Culture preparation of human CD4$^+$ T cell blasts.** The use of the material of human origin in this work was approved by the ethics committee of the Technical University of Munich (approvals 934/03 and 1872/07) and was in accordance with

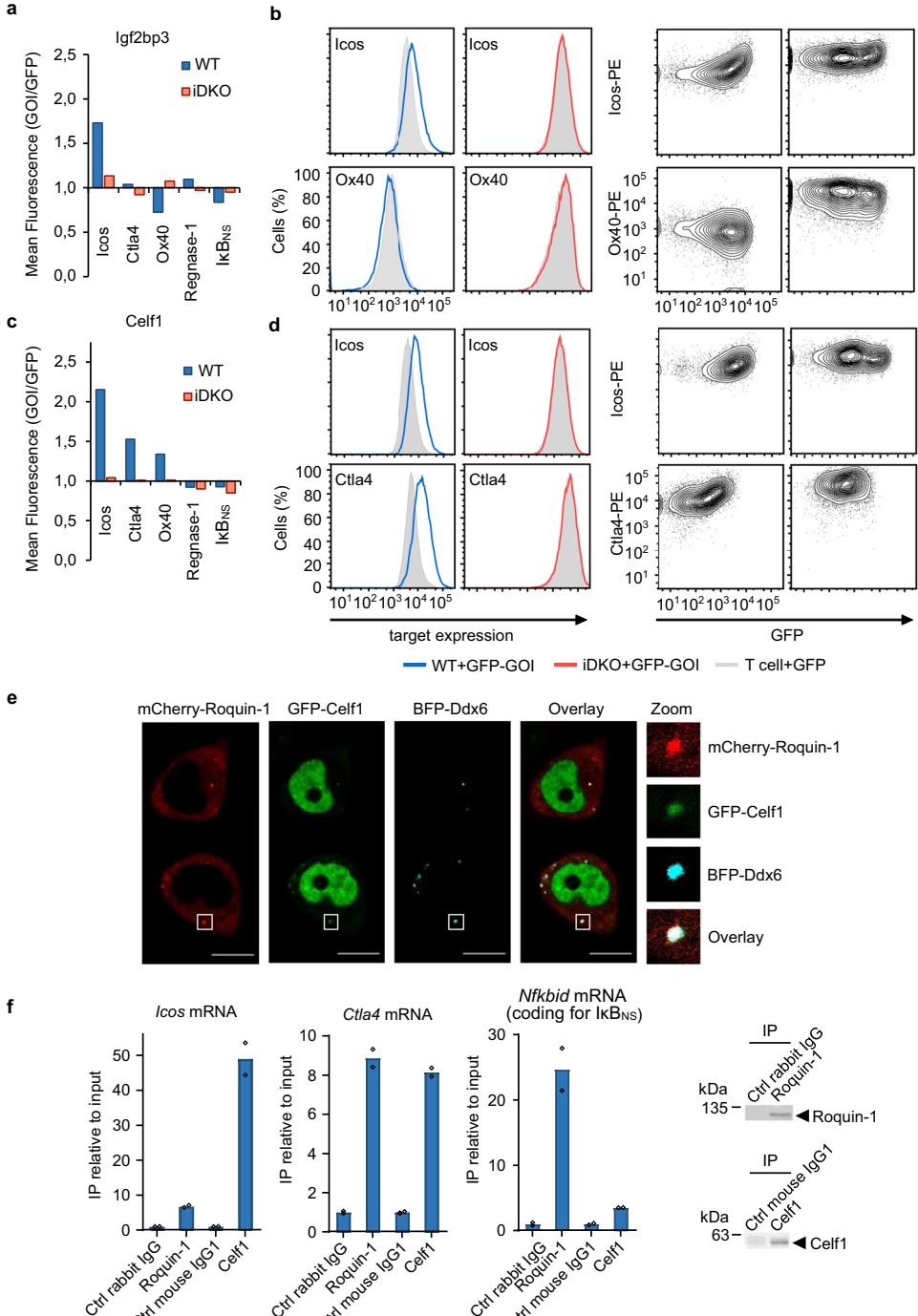

**Fig. 7 Higher-order Icos regulation by Roquin-1 and Celf1.** Inducible overexpression of candidate RBPs in CD4+ T cells was used to investigate cooperative or antagonistic effects on Roquin-1/2-regulated targets. CD4+ T cells with the genotypes *Rc3h1*fl/fl;*Rc3h2*fl/fl;*rtTA3* without (WT) or with the *Cd4-Cre-ERT2* allele (iDKO) were used for transduction with retroviruses after treatment with 4'-OH-tamoxifen. Expression levels of Icos and four additional Roquin-1 targets in WT and iDKO cells were analyzed 16 h after doxycycline-induced arrayed expression of 46 individual GFP-GOI fusion genes. **a, c** Each geometric mean of the Roquin-1 target in the GOI-GFP sample was divided by the geometric mean of the Roquin-1 target in the GFP sample. Summarized ratios are shown as bar diagrams for (**a**) lgf2bp3 and (**b**) Celf1. **b, d** Representative flow cytometry data are depicted as histograms and contour plots. Experiments were repeated at least three times. **e** Using confocal microscopy, co-localization of Roquin-1 and Celf1 in P bodies was demonstrated by co-transfecting Hela cells using plasmids coding for Cherry-Roquin-1, GFP-Celf1, and BFP-Ddx6, the latter of which served as a P body marker (*n* = 2). The standard bars equal 10 μm. **f** For RNA-immunoprecipitation CD4+ T cells were isolated and cultured as for (**a, c**). Anti-Roquin-1, anti-Celf1, and isotype control antibodies were coupled to magnetic beads and used to immunoprecipitate the respective proteins and their bound RNAs from T cell lysates. At the end, Trizol was added to the magnetic beads to isolate RNA, reverse-transcribed cDNA, and measure target mRNA levels by qPCR and relative to input. Isotype levels were set to 1. Depicted is one representative experiment (*n* = 2) and data points indicate technical replicates of qPCR measurements. Source data are provided as a Source Data file.

the Declaration of Helsinki of the World Medical Association (last amended in 2013). Blood (120 ml) was collected by venepuncture from two times four donors for RNA-IC and OOPS. Peripheral blood mononuclear cells (PBMCs) were isolated by standard Ficoll gradient (P04-601000, Pancoll) centrifugation and CD4$^+$ T cells were isolated from $1 \times 10^8$ cells using CD4$^+$ Microbeads (130-045-101, Miltenyi)) to arrive at $2$–$3 \times 10^7$ cells. The purity was >95% CD4$^+$ T cells with >90 viable cells. These were resuspended at $2 \times 10^6$/ml in T cell medium (AIM-V (12055-091, Invitrogen), 10% heat-inactivated human serum, 2 mM L-glutamine, 10 mM HEPES and 1.25 μg/ml Fungizone), supplemented with 500 ng/ml PHA (C03HA16GB, Murex) and 100 IU IL-2/ml (Proleukin S, Novartis). Cells were dispensed in 24-well plates at 2 ml per well and on day 3 old medium was replaced by a fresh T cell medium and cells were harvested and counted on day 4.5. For generating iTreg, naïve CD4$^+$ T cells were isolated from PBMCs using Microbeads (Naive CD4$^+$ T Cell Isolation Kit II, 130-094-131, Miltenyi) and resuspended at $1 \times 10^6$ cells/ml in T cell medium supplemented with 500 ng/ml PHA (Murex), 500 IU IL-2/ml (Proleukin S, Novartis), 500 nM Rapamycin (S1039, Selleckchem), and 5 ng/ml TGF-ß1 (130-095-067, Miltenyi). Cells were cultured in 24-well plates at 2 ml per well together with $1 \times 10^6$ irradiated (40 Gy) PBMCs derived from three donors. On day 3, the old medium was replaced by a fresh T cell medium including supplements, and the cells were harvested and counted on day 5.

**Functional analysis of human CD4$^+$ T cells.** To test for T$_H$1 cytokine production, aliquots of effector CD4$^+$ T cells (Teff) were cultivated for additional 2–3 weeks. Subsequently, $1 \times 10^5$ of these resting Teff cells were washed, resuspended in 200 μl T cell medium and either left untreated or treated with 1.5 μg/ml of the αCD3 antibody OKT3 (Janssen-Cilag). Next day, the IFNγ content in the culture supernatant was measured by ELISA (3420-1H-20, Mabtech). Inhibitory activity of human iTreg cells was assessed in co-culture experiments by adding iTreg cells at different ratios to the co-cultures of $5 \times 10^4$ Epstein-Barr virus-specific CD4$^+$ T cells BALF4-B5[64] and $5 \times 10^4$ autologous EBV-positive lymphoblastoid cells (LCL) in 200 μl of AIM-V medium. IL-2 secretion by the T cells was measured 24 h later by ELISA (130-095-067, Mabtech).

**RBPome capture.** For human or mouse RNA-IC experiments three biological replicates were generated using $20 \times 10^6$ T$_{eff}$ or iTreg cells from 6 to 8 weeks old mice or healthy young human donors. Cells were either lysed directly (non-irradiated, control) in 1 ml lysis buffer from the μMACS mRNA Isolation Kit (130-075-201, Miltenyi) or suspended in 1 ml PBS, dispensed on a 10 cm dish, and UV irradiated with 0.2 J/cm$^2$ at 254 nm, washed with PBS, pelleted and subsequently lysed and mRNA was isolated from both samples with the μMACS mRNA Isolation Kit according to the manufacturer's instructions. RNAs and crosslinked proteins were eluted with 70 °C RNase-free H$_2$O. For RBPome capture for Western blot analysis, $400 \times 10^6$ EL-4 T cells were either lysed directly in 8 ml lysis buffer (nonirradiated, control) or suspended in 16 ml PBS, dispensed on sixteen 10 cm dishes, and UV irradiated at 0.2 J/cm$^2$ at 254 nm for 1 min, washed with PBS, pelleted and subsequently lysed in 8 ml lysis buffer (UV irradiated) and then mRNA was isolated from both samples from the μMACS mRNA Isolation Kit using 500 μl oligo(dT) beads per sample. Each sample was split and run over two M columns (130-042-801, Miltenyi) and each column was eluted with two times 100 μl RNase-free H$_2$O. The eluate was concentrated in Amicon centrifugal filter units (UFC501008, Merck) to a final volume of ~25 μl. 8 μl Lämmli buffer (4×) with 10% (vol/vol) β-mercaptoethanol was added and the sample was boiled for 5 min at 95 °C. For protein analysis, samples were either flash-frozen in liquid nitrogen for MS analysis, or Lämmli loading dye was added for subsequent analysis by Western blotting or silver staining.

**OOPS.** Biological replicates were generated using $20$–$30 \times 10^6$ CD4$^+$ T cells from three 6–8-week-old mice and four healthy, young human donors. CD4$^+$ T cells were activated without cytokines and antibodies that skew their differentiation into specific T helper cell subsets to generate unskewed Teff cells. Cells were washed in PBS once and resuspended in 1 ml of cold PBS and transferred into one well of a six-well dish. Floating on ice, the cells in the open six-well plate were UV irradiated once at 0.4 J/cm$^2$ and twice at 0.2 J/cm$^2$ at 254 nm with shaking in-between sessions. After irradiation, the 1 ml of cell suspension was transferred into a FACS tube and the well was washed with another 1 ml of cold PBS which also added to the FACS tube. After centrifugation, the cell pellet was completely dissolved in 1 ml of TRI Reagent (T9424, Sigma). The remainder of the procedure was performed strictly according to Queiroz et al. [29] with the exception that we broke up and regenerated interphases in five successive rounds of phase partitioning, rather than three as an additional measure of precaution.

**SDS–PAGE, Western blotting, and silver staining.** All protein visualization procedures were performed according to standard protocols. For silver staining we used the SilverQuest kit (LC6070, Invitrogen). The following in-house generated monoclonal antibody supernatants were used in Western blots at a 1:10 dilution: anti-Roquin-1/2, cl. 3F12; anti-Regnase-1, cl. 15D11; anti-pan-Ago, cl. MAGO3-5, anti-Nufip2, cl. 23G8, anti-pan-YTHDF, cl. 17F2; anti-GFP, cl. 3E5-111. Commercial antibodies used for Western blots were: anti-Fxr1, polyclonal, 4173, 1:1000, (Cell Signaling); anti-Fxr2, cl. D85D6, 7098, 1:1000 (Cell Signaling); anti-TTP, cl.

TP6, 1:1000 (Sigma); anti-Gapdh, cl. 6C5, CB1001, 1:10,000 (Calbiochem), anti-Celf1, cl. 850717, MAB9388, 0.5 μg/ml (R&D SYSTEMS); anti-RBMS1, cl. EPR9825(B), ab150353, 1:5000 (abcam); anti-CPEB4, polyclonal, 25342-1-AP, 1:750 (Proteintech); anti-Ptbp1, 1:1000, 8776 (Cell Signaling); anti-tubulin, 1:1000, 86298 (Cell Signaling); goat anti-rat antibody, cl. Poly4054, 1:200 (Biolegend); goat anti-mouse antibody, polyclonal, 554001, 1:400 (BD Bioscience) and anti-Ptbp1, (1:1000, 8776, Cell Signaling). Proteins were visualized by staining with anti-rat (1:3000, 7077, Cell Signaling) or anti-mouse (1:3000, 7076, Cell Signaling) secondary antibodies conjugated to HRP.

**Sample preparation for mass spectrometry.** For RNA-capture, eluates were incubated with 10 μg/ml RNase A in 100 mM Tris, 50 mM NaCl, 1 mM EDTA at 37 °C for 30 min. RNase-treated eluates were acetone precipitated and resuspended in denaturation buffer (6 M urea, 2 M thiourea, 10 mM Hepes, pH 8), reduced with 1 mM DTT, and alkylated with 5.5 mM IAA. Samples were diluted 1:5 with 62.5 mM Tris, pH 8.1, and proteins digested with 0.5 μg Lys-C and 0.5 μg Trypsin at room temperature overnight. The resulting peptides were desalted using stage-tips containing three layers of C18 material (Empore).

For OOPS experiments, 100 μl of lysis buffer (P.O.00027, PreOmics, iST kit) were added and samples were incubated at 100 °C for 10 min at 1400 rpm. Samples were sonicated for 15 cycles (30 s on/30 s off) on a bioruptor (Diagenode). Protein concentration was determined using the BCA assay and about 30 μg of proteins were digested. To this end, trypsin and Lys-C were added in a 1:100 ratio, samples diluted with lysis buffer to contain at least 50 μl of volume, and incubated overnight at 37 °C. To 50 μl of the sample, 250 μl isopropanol/1% TFA was added and samples vortexed for 15 s. Samples were transferred on SDB-RPS (Empore) stagetips (3 layers), washed twice with 100 μl isopropanol/1% TFA, and twice with 100 μl 0.2% TFA. Peptides were eluted with 80 μl of 2% ammonia/80% acetonitrile, dried on a centrifugal evaporator, and resuspended with 10 μl of buffer A* (2% ACN, 0.1% TFA).

**LC–MS/MS analysis.** Peptides were separated on a reverse-phase column (50 cm length, 75 μm inner diameter) packed in-house with ReproSil-Pur C18-AQ 1.9 μm resin (Dr. Maisch GmbH). Reverse-phase chromatography was performed with an EASY-nLC 1000 ultra-high pressure system, coupled to a Q-Exactive HF Mass Spectrometer (Thermo Scientific) for mouse RNA-capture experiments or a Q-Exactive HF-X Mass Spectrometer (Thermo Scientific) for human RNA-capture, OOPS experiments, and single-shot proteomes in combination with Thermo Q-Exactive HF Tune software (v2.4.0.1824), Thermo Q-Exactive HF-X Tune software (v2.9.0.2982) and Xcalibur (v4.0.27.19). Peptides were loaded with buffer A (0.1% (v/v) formic acid) and eluted with a nonlinear 120-min (100-min gradient for human RNA-capture and OOPS experiments) gradient of 5–60% buffer B (0.1% (v/v) formic acid, 80% (v/v) acetonitrile) at a flow rate of 250 nl/min (300 nl/min for human RNA-capture and OOPS). After each gradient, the column was washed with 95% buffer B and re-equilibrated with buffer A. Column temperature was kept at 60 °C by an in-house designed oven with a Peltier element, and operational parameters were monitored in real-time by the SprayQc software. MS data were acquired using a data-dependent top 15 (top 12 for human RNA-capture and OOPS experiments) method in positive mode. Target value for the full scan MS spectra was $3 \times 10^6$ charges in the $300$–$1650 m/z$ range with a maximum injection time of 20 ms and a resolution of 60,000. The precursor isolation window was set to 1.4 $m/z$ and the capillary temperature was 250 °C. Precursors were fragmented by higher-energy collisional dissociation (HCD) with normalized collision energy (NCE) of 27. MS/MS scans were acquired at a resolution of 15,000 with an ion target value of $1 \times 10^5$, a maximum injection time of 120 ms (60 ms for human RNA-capture and OOPS experiments). Repeated sequencing of peptides was minimized by a dynamic exclusion time of 20 s (30 ms for human RNA-capture and OOPS).

**Raw data processing.** MS raw files were analyzed by the MaxQuant software[65] (version 1.5.1.6 for RNA-capture files and version 1.5.6.7 for OOPS files) and peak lists were searched against the mouse or human Uniprot FASTA database, respectively, and a common contaminants database (247 entries) by the Andromeda search engine[66]. Cysteine carbamidomethylation was set as fixed modification, methionine oxidation, and N-terminal protein acetylation as variable modifications. False discovery rate was 1% for both proteins and peptides (minimum length of 7 amino acids). The maximum number of missed cleavages allowed was 2. Maximal allowed precursor mass deviation for peptide identification was 4.5 ppm after time-dependent mass calibration and maximal fragment mass deviation was 20 ppm. Protein intensities were calculated using the MaxLFQ algorithm, which is based on the pairwise calculation of peptide ratios[67]. "Match between runs" was activated with a retention time alignment window of 20 min and a match time window of 0.5 min for RNA-capture experiments, while matching between runs was disabled for OOPS experiments. The minimum ratio count was set to 2 for label-free quantification.

**Data analysis.** Statistical analysis of MS data was performed using Perseus (version 1.6.0.28). Human RNA-capture, mouse RNA-capture, human OOPS, and mouse OOPS data were processed separately. For all experiments, MaxQuant (V1.5.1.6/

V1.5.1.6.7) output tables were filtered to remove protein groups matching the reverse database, contaminants or proteins only observed with modified peptides. Next, protein groups were filtered to have at least two valid values in either the crosslinked or control triplicate. LFQ intensities were log-transformed (base 2) and missing values were imputed from a normal distribution with a downshift of 1.8 standard deviations and a width of 0.2 (0.25 for OOPS data). For RNA-capture experiments, a Student's T-test was performed to find proteins significantly enriched in the crosslinked sample over the non-crosslinked control (false-discovery rate (FDR) < 0.05). As many proteins were identified specifically in the crosslinked sample at intensities too low to find significant differences compared to the imputed values, we additionally considered proteins only identified in two or three replicates of the crosslinked sample, but never in the non-crosslinked control, as RNA-binding proteins. For OOPS experiments, proteins significantly enriched in a Student's T-tests of the organic phase after RNase-treatment over the same sample of the non-crosslinked control (FDR < 0.05) were considered RBPs.

GO term-enrichment analysis was performed using the clusterProfiler package in R (version 4.1.0) as described in the original publication[68]. The mouse or human proteomes served as background for the respective enrichment analysis. Relative abundance of proteins in the single-shot proteome (Fig. 4a, b) was determined by calculating the logarithm (base 2) of the ratio of the LFQ intensity and the number of theoretical peptides. Relative abundance of proteins significant in the mouse OOPS and RNA-IC is shown in the single-shot proteome of mouse CD4$^+$ T cells measured with the OOPS samples. Relative abundance of proteins significant in the human OOPS and RNA-IC experiments is shown in the single-shot proteome of human CD4$^+$ T cells measured with the OOPS samples or RNA-IC samples, respectively. For the four-way Venn comparison to find RBPs identified in more than one RNA-IC or OOPS experiment, human gene names were converted into their homologous mouse counterparts. Protein group entries containing more than one isoform were expanded for the comparison and subsequently collapsed into one entry again for calculation of the Venn diagram. Multiple gene name entries for different unambiguously identified isoforms were collapsed into the major isoform. This affected six protein groups in the human RNA-IC dataset and three protein groups in the mouse RNA-IC dataset. IDRs were retrieved from the Disorder Atlas[69]. We referred to the LCR-eXXXplorer[70] to obtain low complexity regions in proteins.

**Plasmid construction.** To generate a vector that expresses N-terminally GFP-tagged proteins, we amplified the respective genes from cDNA of T$_{eff}$ or Treg cells, added HindIII and KpnI restriction sites in front of the start codon and cloned them into the pCR8/GW/TOPO® vector. We then used HindIII and KpnI to insert a GFP sequence where we removed the bases for the stop codon. The respective sequences were subsequently transferred to the expression vector pMSCV via the gateway cloning technology. Only GFP-Roquin-1 was expressed from the vector pDEST14. All other cloning was performed using In-Fusion technology (639649, Takara). All oligonucleotide sequences are compiled in Supplementary Table 5.

**Validation of RNA-binding ability.** HEK293T cells were transfected by calcium phosphate transfection with plasmids expressing the respective proteins with an N-terminal GFP-tag or GFP alone. After three days, cells were washed with PBS on plates, UV crosslinked (CL) as before or directly scraped from the plates (nCL). Cell lysates were generated by flash-freezing pellets in liquid nitrogen and incubation in NP-40 lysis buffer (150 mM NaCl, 1% NP-40, 50 mM Tris–HCl, pH 7.4, 5 mM EDTA, 1 mM DTT, 1 mM PMSF and protease inhibitor mixture (04693159001, Complete, Roche)). After lysis, extracts were cleared by centrifugation at 17,000 × g for 15 min at 4 °C. We then determined protein concentration via the BCA method (23227, Thermo Fisher) and used 2–10 mg of protein for the subsequent GFP immunoprecipitation, depending on transfection efficiency and expected RNA-binding capacity. We pre-coupled 200 µl Protein-G beads (10004D, Dynabeads Protein G, Invitrogen) with 20 µg antibody (anti-GFP, clone: 3E5-111, in house) in PBS (1 h, RT), washed beads in lysis buffer, added them to cell lysates and incubated with rotation for 4 h at 4 °C. Beads were then washed three times with IP wash buffer (50 mM Tris–HCl, pH 7.5) with decreasing salt (500, 350, 150, 50 mM NaCl) and SDS (0.05%, 0.035%, 0.015%, 0.005%) concentrations. Proteins and crosslinked RNAs were eluted with 50 mM glycine, pH 2.2 at 70 °C for 5 min. Lämmli buffer (4×) was added and samples were divided for mRNA and protein detection and separated via SDS gel electrophoresis (6% SDS gels for detection of mRNA samples and 9% gels to verify immunoprecipitation efficiency). For RNA detection we blotted onto Nitrocellulose membranes and for protein detection on PVDF membranes. After transfer, the Nitrocellulose membrane was prehybridized with Church buffer (0.36 M Na$_2$HPO$_4$, 0.14 M NaH$_2$PO$_4$, 1 mM EDTA, 7% SDS) for 30 min and then incubated for 4 h with Church buffer containing 40 nM 3′-and 5′-biotin-labeled oligo(dT)$_{20}$ probe to anneal to the poly-A tail of the bound mRNA. The membrane was washed twice with 1× SSC, 0.5% SDS and twice with 0.5× SSC, 0.5% SDS. Bound mRNA was detected with the Chemiluminescent Nucleic Acid Detection Kit Module (89880, Thermo Fisher) according to the manufacturer´s instructions.

**Real-time PCR.** Total RNA of input was purified from lysates with Agencourt RNAClean XP Beads (A63987, Beckman Coulter) according to the manufacturer´s instructions and eluted in nuclease-free H$_2$O. cDNA was synthesized from total

input RNA and oligo(dT)-isolated RNA with the QuantiTect Reverse Transcription Kit (205311, Qiagen). The respective qRT-PCRs for Hprt, β-actin, and 18S rRNA were performed with the SYBR green method. RNA isolation, reverse transcription, and quantitative RT PCR for Renilla and Firefly luciferases, Rc3h1, Icos, Ctla4, Nfkbid, Celf1, and Celf2 were performed as published[71] using the universal probes systems (Roche). For Primer sequences see Supplementary Table 5.

**Expression and purification of Stat proteins.** The pGEX-6P-2/Stat constructs (human STAT1α and STAT4 as well as mouse Stat1 and Stat4) were transformed into E. coli Rosetta2 (DE3) or Rosetta2 (DE3) pLysS and expressed overnight at 20 °C in ZYM 5052 auto-induction medium (3S2000, Teknova). The cells were harvested, lyzed by sonication, and clarified by centrifugation. Non-specifically bound bacterial nucleotides were precipitated by the addition of 0.5% poly-ethylenimine (PEI) and the excess PEI was removed by ammonium sulfate precipitation at 95% saturation. The ammonium sulfate pellet was resuspended in GSTrap-binding buffer, dialyzed overnight against the same buffer and the Stat proteins were then purified on a GSTrap column. Next, the GST-tag was removed by the addition of HRV 3C protease (1:25 molar ratio) and the Stat proteins further purified by size exclusion chromatography over a Superdex 200 column.

**In vitro transcription.** In vitro transcription of the non-coding RNA 'TSU' (AF080092) was performed using the HiScribe T7 High Yield RNA Synthesis Kit (E2050S, NEB) according to the manufacturer's instructions. Subsequently, the template DNA was digested using TURBO DNase (AM2238, ThermoFisher) and the cleanup procedure of the RNeasy Mini kit (74104, Qiagen) was applied to obtain the 481 nt long TSU RNA that was used in EMSA's.

**Electrophoretic mobility shift assay.** Purified Stat proteins (0, 50, and 150 pmol) were combined with in vitro transcribed TSU RNA (1.5 pmol) in EMSA buffer (150 mM NaCl, 50 mM Tris–HCl pH 8.0, 1 mM MgCl$_2$ and 1 mM dithiothreitol). BSA and glycerol were used to adjust for protein concentration and storage buffer conditions. After 20 min on ice, 5 µl of glycerol (30%) were added to the 20 µl reactions and 20 µl were subsequently loaded and run on 1.5% agarose gels. Gel documentation was carried out after staining in an ethidium bromide bath.

**Tethering assay.** Hela cells were seeded in 24-well plates using 5 × 10$^4$ cells per well. Transfection was performed the following day using Lipofectamine2000 (11668019, Thermo Fisher) and 300 ng of total constructs. Each transfection consisted of 75 ng of luciferase reporter plasmid psiCHECK2 (C8021, Promega) or luciferase-5boxB plasmid psiCHECK2 -5boxB, 225 ng of pDEST12.2-λN fused constructs. After 24 h, cells were harvested for luciferase activity assays using a Dual-Luciferase Reporter Assay System (E1910, Promega). Renilla luciferase activity was normalized to Firefly luciferase activity in each well to control for variation in transfection efficiency. psiCHECK2 lacking boxB sites served as a negative control, and each transfection was analyzed in triplicates.

**BioID.** The proximity-dependent biotin identification assay was performed according to Roux[57] with modifications. For each sample 2 × 10$^7$, MEF cells were grown on ten 15-cm cell culture dishes for 24 h before BirA*-Roquin-1 or BirA* expression was induced by doxycycline treatment (1 µg/ml). For T cells, transduction with the same BirA*-fusions cloned into the plasmid pRetroXtight was performed as described above and the same number of cells was used for the experiment. Six hours after the addition of doxyclycline, biotin was added for 16 h to arrive at an end concentration of 50 µM. Approximately 8 × 10$^7$ cells per sample were trypsinized, washed twice with PBS, and lysed in 5 ml lysis buffer (50 mM Tris–HCl, pH 7.4; 500 mM NaCl, 0.2% SDS; 1× protease inhibitors (04693159001, Roche), 20 mM DTT, 25 U/ml Benzonase (1.01654.0001, Merck)) for 30 min at 4 °C using an end-over-end mixer. After adding 500 µl of 20% Triton X-100 the samples were sonicated for two sessions of 30 pulses at 30% duty cycle and output level 2, using a Branson Sonifier 450 device. Keep on ice for 2 min in between sessions. Pipetting of 4.5 ml prechilled 50 mM Tris–HCl, pH 7.4 was followed by an additional round of sonication. During centrifugation at 16,500 × g for 10 min at 4 °C, 500 µl Dynabeads MyOne Streptavidin C1 (65002, Invitrogen) or each sample was equilibrated in a 1:1 mixture of lysis buffer and 50 mM Tris–HCl pH 7.4. After overnight binding on a rotator at 4 °C Streptavidin beads were stringently washed using wash buffers 1, 2, and 3 ([57]) and prepared for mass spectrometry by three additional washes with buffer 4 (1 mM EDTA, 20 mM NaCl, 50 mM Tris–HCl pH 7,4). Proteins were eluted from streptavidin beads with 50 µl of biotin-saturated 1× sample buffer (50 mM Tris–HCl pH 6.8, 12% sucrose, 2% SDS, 20 mM DTT, 0.004% Bromphenol blue, 3 mM Biotin (B4501, Sigma) by incubation for 7 min at 98 °C. For identification and quantification of proteins, samples were proteolysed by a modified filter aided sample preparation[72], and eluted peptides were analyzed by LC–MSMS on a QExactive HF mass spectrometer (ThermoFisher Scientific) coupled directly to a UItimate 3000 RSLC nano-HPLC (Dionex). Label-free quantification was based on peptide intensities from extracted ion chromatograms and performed with the Progenesis QI for proteomics v3.0 software (Nonlinear Dynamics, Waters). Raw files were imported and after alignment, filtering, and normalization, all MSMS spectra were exported and searched against the Swissprot mouse database (16772 sequences, Release 2016_02) using the Mascot search

engine (Version 2.5.1 and 2.6.1) with 10 ppm peptide mass tolerance and 0.02 Da fragment mass tolerance, one missed cleavage allowed, and carbamidomethylation set as fixed modification, methionine oxidation, and asparagine or glutamine deamidation allowed as variable modifications. A Mascot-integrated decoy database search calculated an average false discovery of <1% when searches were performed with a mascot percolator score cut-off of 13 and a significance threshold of 0.05. Peptide assignments were re-imported into the Progenesis QI for proteomics v3.0 software. For quantification, only unique peptides of an identified protein were included, and the total cumulative normalized abundance was calculated by summing the abundances of all peptides allocated to the respective protein. A t-test implemented in the Progenesis QI software comparing the normalized abundances of the individual proteins between groups was calculated and corrected for multiple testing resulting in q values (FDR adjusted p values) given in Supplementary Data 2. Only proteins identified by two or more peptides were included in the list of Roquin-1 proximal proteins.

**Confocal microscopy.** One day prior to analysis, HeLa cells were transfected via calcium-phosphate precipitation, and cells were seeded 6 h prior to microscopic analysis on eight-well μ-slides (Glass bottom, Ibidi) in Leibovitz's L-15 media (no phenol red, 21083027, Thermo Fisher). Confocal images were performed with a TCS SP8 X FALCON confocal head (Leica Microsystems, Wetzlar, Germany) mounted on an inverted microscope (DMi8; Leica Microsystems). For confocal imaging, a 405 nm diode and a white light laser were used as excitations sources (405 nm for BFP, 488 nm for GFP, and 594 nm for mCherry). Single photons were collected through a ×93/1.3 NA glycerin-immersion objective and detected on Hybrid Detectors (HyD) (Leica Microsystems) with a 414–468, 500–550, and 610–722 nm spectral detection window for BFP, GFP, and mCherry detection, respectively. The image size was set to 512 × 512 pixels and a 2.5-fold zoom factor was applied, giving a pixel size of 0.098 μm and an image size of 50 × 50 μm. Scanning speed was 600 Hz.

**Co-immunoprecipitation of Roquin or Celf1 bound RNAs.** CD4 T cells ($3 \times 10^7$) were lysed on ice in 500 μl cold, RNase-free lysis buffer (20 mM Tris, pH 7.5, 150 mM NaCl, 0.25% (vol/vol) Nonidet-P40, 1.5 mM MgCl$_2$, protease inhibitor mix without EDTA (04693159001, Roche) and 1 mM dithiothreitol). Lysates were shock frozen, thawed, and cleared by centrifugation (10 min, 12,000 × g, 4 °C). Rabbit anti-roquin (A300-514A; Bethyl, 6 μg) and mouse monoclonal anti-Celf1 (ab9549; Abcam, 2 μg) were bound to 50 μl protein A (10001D, Invitrogen) or G (10004D, Invitrogen) magnetic beads, respectively, and incubated for 1 h at 4 °C. In parallel, the protein lysate was pre-cleared using 20 μl of uncoupled magnetic beads. Cleared lysates and antibody-coupled beads were combined, incubated for 4 h at 4 °C, and then washed with lysis buffer four times. Finally, 1 ml of TRI reagent (T9424, Sigma) was added to the magnetic beads, and RNA was purified according to the standard protocol. Quantitative PCR was performed on immunoprecipitated RNA and input, using UPL probes (Roche) and the primers listed in Supplementary Table 5.

**Antibodies.** To generate monoclonal antibodies against pan-Ythdf proteins or λN-peptide, Wistar rats were immunized with purified GST-tagged full-length mouse Ythdf3 protein or an ovalbumin-coupled λN peptide (MNARTRRRER-RAEKQWKAAN) using standard procedures as described[73]. The hybridoma cells of Ythdf- or λN-reactive supernatants were cloned at least twice by limiting dilution. Experiments in this study were performed with anti-pan-Ythdf clone DF3 17F2 (rat IgG2a/κ) and anti-λN clone LAN 4F10 (rat IgG2b/κ).

**Reporting summary.** Further information on research design is available in the Nature Research Reporting Summary linked to this article.

## Data availability

The mass spectrometry proteomics data that support the BioID results have been deposited to the ProteomeXchange Consortium via the PRIDE[74] partner repository with the dataset identifier PXD026716. The mass spectrometry data that support the identification of the CD4$^+$ T cell RBPome have likewise been deposited to PRIDE with the accession codes PXD008830 (mouse RNA-IC), PXD021164 (human RNA-IC), PXD022795 (mouse OOPS), PXD021169 (human OOPS). Source data are provided with this paper.

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

## Acknowledgements

We thank Hemalatha Mutiah (Ludwig Maximilians University Munich) for screening hybridoma supernatants and also Claudia Keplinger (Helmholtz Center Munich) and Mariano Gonzalez Pisfil (BMC/LMU core facility bioimaging) for excellent technical support. For the provision of mouse lines with floxed alleles we would like to thank Marc Schmidt-Supprian (Rc3h1), Wolfgang Wurst (Rc3h2), Robert Blelloch (Dgcr8), Mingui Fu (Zc3h12a) and Thorsten Buch (Cd4-Cre-ERT2). Artwork was produced by K.P.H. The work was supported by the German Research Foundation grants SPP-1935 (to V.H.), SFB-1054 projects A03 and Z02 as well as HE3359/7-1 and HE3359/8-1 to V.H. as well as grants from the Wilhelm Sander, Fritz Thyssen, Else Kröner-Fresenius and Deutsche Krebshilfe foundations to V.H. D.B. was supported by Deutsche Forschungsgemeinschaft (DFG, German Research Foundation) under Emmy Noether Program BA 5132/1-1 and BA 5132/1 2 (252623821), as well as SFB 1054 project B12 (210592381) and Germany's Excellence Strategy EXC2151 (390873048).

## Author contributions

E.G. and V.H. conceived the idea for the project and together with M.W. and M.M. supervised the experimental work. K.P.H. and V.H. wrote the manuscript with contributions from E.G., M.W., and A.R. K.P.H. performed OOPS, BioID, and T cell transductions with help from G.B., K.D., S.M.H., J.M. and C.C. C.G. conducted RNA-IC experiments and RNA-binding assays. A.R. and S.M.H. performed mass spectrometry and A.R. analyzed RBPome data. S.M.H. and K.P.H. analyzed the BioID data. M.X. contributed the tethering assays, L.K. the confocal microscopy and E.H.W. analyzed Icos regulation for which A.G., R.F., T.I.-K. and D.B. provided unpublished reagents.

## Funding

## Competing interests

The authors declare no competing interests.
