## [Peer Review File · Nature Communications]

Defining the RBPome of primary T helper cells to elucidate higher order Roquin-mediated mRNA regulationREVIEWER COMMENTS

Reviewer #1 (Remarks to the Author):

The manuscript "Defining the RBPome of T helper cells to study higher order post-transcriptional gene regulation" by Hoefig, Reim and Gallus et al. report the cataloguing of the human and mouse T helper cell RBPome obtained from primary cells expanded in vitro and enriched through the classic oligo-d(T) affinity and the novel organic phase separation methods. They validate the RBP status of a few putative RBPs enriched in their samples and also follow up with a functional study of a subset of RBPs affecting the post-transcriptional control of the Icos mRNA.

The study is mostly well designed and the use of primary cells make it especially relevant. Unfortunately, the manuscript is poorly written with imprecise descriptions and lack of some essential detail, which is disappointing. An overhaul of the manuscript is thus necessary before it is suitable for publication in Nature Communications.

1) In general, the experimental design is poorly described with only a mention in passing that replicates were performed, but the extent or type of replication is not described. It is not clear in the text, for example, how many replicates (different donors) were enriched with each method or after how many days post-expansion the cells were harvested and which state they were at (viability, activation phenotype, percentage of different subpopulations present...). The number of donors (n) should be reported in the figure legends as well.

2) The supplementary tables were not available to review! This is a woeful omission and does not inspire the reader to have any confidence in the results presented. The tables should be compiled for data post processing for both methods are made available as well as raw quantitation output from search engine (without imputation). That will be useful for groups that wish to process this data using their preferred tools.

3) All raw mass spectrometry files and raw search engine output must be made available in PRIDE or MASSIVE repositories with passwords in place for the reviewers. This is also an unfortunate oversight.

4) What justifies the use of Match between runs with RNA-IC but not with OOPS? I strongly question the reliability of match between runs, especially in affinity purifications. The data must be treated equally in both analysis and, if match between runs is used, I would request supplementary information discussing clearly how much of the data is based in "matched" information and the data without imputation or matching should also be available.

5) LFQ is usually normalised across conditions/replicates by loading equal amounts per run. How were the CL and nCL conditions normalised for fair comparison since nCL inherently recovers much less amount and diversity of proteins? This is a crucial point to establish reliability of the RBPome catalogue and, thus, must be clearly explained and demonstrated in the main text.

6) The authors do not make clear why they used two separate RNA binding protein enrichment methods. OOPS gave more data than RNA-IC. If the authors considered that RNA-IC might give deeper coverage of the mRNA binding proteome than OOPS, then this should be clearly articulated. As RNA-IC does not exclusively enrich polyadenylated RNA, some clear explanation should be given. The reader at the moment can only assume that the two methods were used sequentially and OOPS possibly being an add on as a newer method.

7) Line 175-177 states that difference in the RNA-IC derived RBPome for different types of T cells could have arisen from an incomplete assessment of the RNA-IC methods. I don't follow this argument as the same method was used for all - it would suggest the replicates are variable rather than incomplete assessment.

8) I would like to have seen a fuller assessment of the types of proteins enriched by RNA-IC method but not OOPS. Given OOPS is more sensitive overall, perhaps proteins enriched by RNA-IC over OOPS have specific physical properties and/or biological roles.

9) In the methods section, the authors cross-link Teff or iTreg cells at 0.2 J/cm² at 254nm for 1 min in the RNA-IC section. In the OOPS section, they claim that the cells 'floating on ice' were irradiated once at 0.4 J/cm² and twice at 0.4 J/cm² at 254nm (for how long?). There is no mention of why these two different sets of parameters were used, why the CD4+ T cells required additional cross-linking or how these parameters were optimized. This is crucial, as over irradiation may lead to additional stress responses in the system, and insufficient cross-linking will lead to inefficient capturing of the RBPome. Moreover, in the OOPS methods section the authors carry out five successive rounds on phase partitioning, with no mention of why they chose this number and what benefit additional rounds over and above what reported in the original OOPS paper (Queiroz

et Nat. Biotech 2019) was observed. Given that UV crosslinking needs to be optimized for each cell line and there is limited data for UV crosslinking efficiencies for T cell RBPs, I was surprised that no optimization data was proffered in this manuscript.

Minor points

Line 37: 'including Roquin' – you should be clear whether you are talking about Roquin 1 or 2 or both

Lines 49-53 would benefit with some additional references

Line 55: Previous studies in T helper cell

Line 64: Which Roquin binds together with Nufip2 to RNA?

Line 75. It is false to claim RNA-IC identifies "exclusively" poly-A RNAs since it also identifies ribosomal proteins for instance as clearly observed in published studies.

Line 77: Where is the original OOPS citation?

Line 105: Which microRNAs repress Roquin 1 and 2. Define a TTP binding site?

Line 143: "To decipher the post-transcriptional the network" . remove "the" before "network"
Sections "The polyadenylated RNA-bound proteome of mouse and human T helper cells" and "the global RNA-bound proteome of mouse and human T helper cells" : the number of cells and their state should be reported in the main text

Line 180: Insert OOPS citation

Line 211: 'In comparison to single shot total proteomes'...not clear what is mean by single shot total proteomes – no replicates? single injection on LC-MS/MS??

Line 224: what is meant by (comprises 26 to 224 members)? I assume the authors mean RNA binding domains, but does the number refer to different domains or the total number of proteins? For example, a protein might have two KH domains, so is this counted as 1 or 2 members?

Line 227: 'These findings underscored that the OOPS method recovered RBPs from additional, non-polyadenylated RNAs and that RNA-IC-derived RBPomes are specific but incomplete'. This is inaccurate...the OOPS method is more efficient and may therefore also capture additional mRNA interacting proteins and RNA-IC derived RBPomes are not completely specific as reported in many previous RIC publications.

Line 235: what do the authors mean by false positives? Proteins that partition to the interphase during the OOPS protocol and are released by RNase treatment despite not binding RNA? Proteins that bind indirectly to RNA binding proteins during both enrichment protocols used?

Line 285: if Roquin -1 binds RNA in one part of the cell and not in elsewhere how do you know are that your interaction list is only involved with the Roquin 1 RBP interaction complex?

Line 293: What is meant by 'Reflecting deletion'? This is poor scientific language

Line 322: Atlas would suggest some spatial information - as total cell lysates were used here, what the authors have created is a list not an atlas.

Line 333: what do they mean by "false-positive proteins"? do the authors believe that the controls used are not adequate?

Line337: define and justify 'cut-off'

Line 354-355: This sentence does not make sense

Line 415: 'squished' is not an especially precise scientific term!

Line 518-9: What do the authors mean when they say 'activated without bias'?

Line 621: 'width of 0.2 (0.25 for OOPS data).' What does this mean and how can the authors use different parameters for the two different datasets – where is the statistical justification?

Reviewer #2 (Remarks to the Author):

Hoefig et al. present a comprehensive study shows that Icos responds from several RBPs, including Roquin, Regnase-1, Wtap, Dgcr8. They found that hundreds RBPome of mouse and human T cell proteins by RNA-IC and OOPS. Interestingly, they also found STAT1, STAT4 as the core proteome unexpectedly. Also, they identified RBPs which are related with Roquin-1 in CD4+ T cells. The paper is clearly written and technically sound. There are no ethical concerns arising. The methods are properly conducted, statistical analysis of data is sound and claims are appropriately discussed. The authors' claims are mostly convincing.

I have only some minor suggestions below.

1. I did not see any tables (table 1, supplementary table 1, 2, 3). Please make sure that.
2. In figure 1c, f, i, l, I can see decreased each protein level, but I cannot judge deletion efficiency. It is better to show with immunoblot.
3. In figure 1, Did you see any changes phenotype of T cells by deletion of Roquin, Regnase, Wtap, or Dgcr8?
4. Why did you used Th1 subset for this study, not others? If you use other subset, do you think you will see same result?
5. What is definition of Teff cells? Please clarify.
6. In line 427, "~ initial cell density of 5 or 1.5x10⁶ cells/ml" why did authors culture in 5x10⁶ cells/ml? It is not usual. In that high concentration, cells cannot be activated properly. Does it mean 0.5? Please explain that.

Reviewer #3 (Remarks to the Author):

In this study, the authors identified the core RBPome in mouse and human in T cells by performing both oligo-dT capture and orthogonal organic phase separation (OOPS). The authors found STAT1, STAT4, and Vav1, proteins that had been characterized as transcription factors (TFs) or a guanine nucleotide exchange factor, in the core RBPome, and the tethering assay revealed that these proteins are potent to downregulate gene expression by directly binding with RNA. Then the authors tried to identify Roquin binding proteins comprehensively by performing BioID experiments, which resulted in the identification of about 50 proteins as RBPs in proximity to Roquin-1. Among them, the authors found that a set of RBPs, including Rbms1, Cpeb4, Igf2bp3, and Celf1, could modulate the expression of a set of Roquin-1 target mRNAs. Collectively, the authors conclude that the different proteomic approaches could determine higher-order functional RNA-protein interactions.

Overall, this manuscript is largely descriptive and preliminary due to the lack of mechanistic analysis for the functions of newly identified RBPs. This manuscript is comprised of two parts, the identification of RBPome in T cells and the identification of Roquin-associating proteins, though the relationship between these two stories is not clear. It is rather recommended to separate them into two different manuscripts.

Specific comments are shown as follows:

Major comments

1. In Figure 1, the authors argue that there is a coordinated higher-order regulation by several RBPs, focusing on Icos expression. Although Roquin and Regnase-1 had been shown to directly regulate Icos expression, it is not clear if the lack of Wtap or Dgcr8 directly controls Icos mRNA via m⁶A methylation or microRNAs. Alternatively, the impaired activation of T cells due to the lack of the genes resulted in the altered expression of Icos. Furthermore, there is no evidence that RBPs listed in Fig. 1m-q directly control Icos expression. Thus, the data are not sufficient to claim that there are higher-order post-transcriptional regulation just by monitoring the expression change in a set of RBPs and Icos upon T cell activation
2. The authors determined RBP profiles in mouse and human T cells, which resulted in the identification of 300 and 1000 RBPs by oligo-dT and OOPS methods, respectively (Figures 2-4). Although this is a major part of this manuscript, the authors just analyzed GO enrichment or protein recovery. It is interesting to compare the RBPome of different cell types to analyze if the RBPome contains cell type-specific RBPs.
3. In Figure 5, based on their RBPome data, the authors identified Stat1 and Stat4 as potential RNA-binding proteins. Since UV crosslink at 254 nm can irreversibly bridge the interaction between proteins and nucleic acids, DNA is possibly contaminated in this assay, resulting in the identification of DNA-binding proteins (Conrad T., et al., Nature Communications 2016, Perez-Perri J., et al., Nature Communications 2018). To exclude this possibility, the authors need to show more direct evidence as to the interaction of these Stat proteins with mRNA such as the EMSA assay by using recombinant proteins.
4. The authors tethered Stat proteins to the 3' UTR of luciferase mRNA by using the λ N-BoxB system, and claim that Stat1 and Stat4 harbor RNA regulatory functions. However, the system is artificial and preliminary. First, it is not clear if Stat1 and Stat4 regulate endogenous mRNAs.

Second, the authors need to clarify how Stat1 and Stat4 suppress protein production. Do they induce degradation of mRNA or inhibit protein translation?

5. It is also interesting and important to examine if the stimulation of T cells with IFN or IL-12 alters the Stat-mRNA binding capacity and the activity to suppress protein production.

6. Do other Stat proteins also function as the RBP?

7. In Figure 6, the authors took advantage of the Bio-ID system to identify proteins in proximity to Roquin-1. However, data shown in Figures 6 and 7 are independent of the data in Figures 1-5.

Thus, it is rather better to publish them as two distinct studies.

8. The BioID experiment identified only 64 and 143 Roquin-1 proximity proteins in primary CD4 T cells and fibroblasts, respectively. Surprisingly, the lists are not so overlapping between T cells and fibroblasts. Does this mean the Roquin-1 complex differs depending on the cell type? Or is this because the experiments did not reach saturation? The proteins checked by the authors in Figure 1m seems to be detected only in MEFs by the BioID assay. The authors need to explain the reason why they focused on these proteins.

9. The authors examined the localization of the candidate protein transfected with the corresponding GFP-fused constructs in HEK293T cells (Supplementary Fig 7c). However, they failed to show the localization of Roquin-1 in the same cells to claim the close proximity of the Roquin-1 to the protein of interest.

10. In Figure 7, the authors chose four RBPs (i.e. Rbms1, Cpeb4, Igf3bp2, and Celf1) out of 46 candidate proteins, and tested if these RBPs can modulate Roquin-targeting mRNAs under wild-type and Roquin-deficient conditions. They finally concluded that there are higher-order functional interactions that is Roquin-dependent. However, the data are so preliminary and descriptive, and they are not so informative without further detailed analysis. For instance, they show that Rbms1 expression upregulated Ctla4, but not some of the other Roquin-1 target mRNAs, irrespective of the presence of Roquin proteins. However, this study lacks the mechanistic analysis of how Rbms1 specifically controls Ctla4 expression. And also, it is not clear the reason for the presence of Roquin-1 in close proximity to Rbms1. Similarly, the authors showed that the expression of Igf2bp3 and Celf1 downregulated the expression of Roquin-targeting genes in wild-type, but not in Roquin-deficient cells. However, they failed to show the relationship between Roquin and these proteins.

11. Supplementary Tables 1-3 are missing in manuscript files.

Minor comments

1. In Figure 1(c, f, I, and l), the efficiency of gene deletion was confirmed by flow cytometry, but negative control was missing. Especially, the deletion efficiency for Regnase-1 looks very marginal. Western blot analysis may solve the problem of high background.

2. Figure legends for Figure 1 should be described precisely. The authors failed to indicate how T cells were stimulated in Figure 1 (b, e, h, and k). They described that Icos expression declined after the "removal" of the TCR stimulus, which makes data confusing as well. There is no statement on this in the Figure legend, although it was described for Figure 1m-q.

3. In Figure 5a and 5b, the authors performed pull-down assays using an anti-GFP antibody, followed by western and northern blot analysis by using oligo(dT) probe. If so, the northern blotting should identify mRNAs with various length, and the blots will give smear bands if they bind with RNA without high specificity. However, the Figures indicate that there is clear "band" of mRNA following capture. Does this mean the RBPs shown in Figures 5a and 5b interact with specific mRNAs?

4. In page 10 (line 245), typos North-Western blot.

Point-by-point response

We would like to thank all reviewers for their interest, insights, comments and their help in improving our manuscript.

*We have performed all the requested new experiments and included them as figures (new **Fig. 4a**, **Fig. 5c** and **5h**, **Fig. 7e** and **7f**, **Supplementary Fig. 1a**, **Supplementary Fig. 2a – 2d** and **Supplementary Fig. 3a**) in the new version of our manuscript. Because of the multitude of new results we had to rearrange some of the figures and add **Supplementary Fig. 9**. Former **Fig. 7e – 7h** became new **Supplementary Fig. 8e – 8h**. **Supplementary Fig. 9** includes new **Fig. 9a and 9d**, and former **Supplementary Fig. 8e and 8f** became new **Supplementary Fig. 9b and 9c**. Moreover, we have addressed all the points raised as detailed below.*

Reviewer #1 (Remarks to the Author):

The manuscript "Defining the RBPome of T helper cells to study higher order post-transcriptional gene regulation" by Hoefig, Reim and Gallus et al. report the cataloguing of the human and mouse T helper cell RBPome obtained from primary cells expanded in vitro and enriched through the classic oligo-d(T) affinity and the novel organic phase separation methods. They validate the RBP status of a few putative RBPs enriched in their samples and also follow up with a functional study of a subset of RBPs affecting the post-transcriptional control of the Icos mRNA.

The study is mostly well designed and the use of primary cells make it especially relevant. Unfortunately, the manuscript is poorly written with imprecise descriptions and lack of some essential detail, which is disappointing. An overhaul of the manuscript is thus necessary before it is suitable for publication in Nature Communications.

We thank this reviewer for appreciating our study design and use of primary cells. We have improved the descriptions according to the criticisms.

1) In general, the experimental design is poorly described with only a mention in passing that replicates were performed, but the extent or type of replication is not described. It is not clear in the text, for example, how many replicates (different donors) were enriched with each method or after how many days post-expansion the cells were harvested and which state they were at (viability, activation phenotype, percentage of different subpopulations present...). The number of donors (n) should be reported in the figure legends as well.

To address this criticism, we now show representative flow cytometry analyses that accompanied the preparation of mouse or human CD4⁺ T cells (**Supplementary Fig. 2a-c**) to reveal purity, phenotype as well as the functional interactions of effector T cells with regulatory T cells (**Supplementary Fig. 2d**). In addition, we now also provide a statement on the preparation of human T cells in the materials and methods section: "The purity was >95% CD4⁺ T cells with >90 viable cells." The number of donors and mice for biological replicates are indicated in the text and Figure legends. We also provide a paragraph on the functional evaluation of human T cells:

Functional analysis of human CD4⁺ T cells

To test for Th1 cytokine production, aliquots of effector CD4⁺ T cells (Teff) were cultivated for additional 2-3 weeks. Subsequently, 1x10⁵ of these resting Teff cells were washed, resuspended in 200 μ l T cell medium and either left untreated, or treated with 1.5 μ g/ml of the α CD3 antibody OKT3 (Janssen-Cilag). Next day, the IFN γ content in the culture supernatant was measured by ELISA (Mabtech). Inhibitory activity of human iTreg cells was assessed in co-culture experiments by adding iTreg cells at different ratios to the co-cultures of 5x10⁴ Epstein-Barr virus-specific

CD4⁺ T cells BALF4-B5⁶⁴ and 5x10⁴ autologous EBV-positive lymphoblastoid cells (LCL) in 200 µl of AIM-V medium. IL-2 secretion by the T cells was measured 24h later by ELISA (Mabtech)."

The wording of the description of mouse T cell cultures in the materials and methods section has been marginally adjusted to replace the word 'squished' and to clarify the number of cells that were used per well (reviewer 2)." The Figure legends of Fig. 2 and Fig.3 now contain the information: "T_H0 cultures from 3 mice or 3 human donors were used as biologic replicates to determine proteins interacting with mRNA."

2) The supplementary tables were not available to review! This is a woeful omission and does not inspire the reader to have any confidence in the results presented. The tables should be compiled for data post processing for both methods are made available as well as raw quantitation output from search engine (without imputation). That will be useful for groups that wish to process this data using their preferred tools.

We are deeply sorry for neglecting to include the supplementary tables in the first submission. However, it was an honest mistake, as evident from us making these Excel files freely available on bioarchives (<https://www.biorxiv.org/content/10.1101/2020.08.20.259234v1.supplementary-material>) on August 20th, 2020.

3) All raw mass spectrometry files and raw search engine output must be made available in PRIDE or MASSIVE repositories with passwords in place for the reviewers. This is also an unfortunate oversight.

We apologize. The mass spectrometry data that support the RBP identification by RNA-IC and OOPS in CD4 T cells had been deposited at PRIDE with the accession codes PXD022795, PXD021169, PXD021164, PXD008830. We already included these accession numbers in the accompanying reporting-summary of our initial submission, but by accident we did not include the reviewer passwords. The accession numbers are now also present in the data availability statement. Here are the reviewer usernames and passwords for PRIDE:

[Redacted]

4) What justifies the use of Match between runs with RNA-IC but not with OOPS? I strongly question the reliability of match between runs, especially in affinity purifications. The data must be treated equally in both analysis and, if match between runs is used, I would request supplementary information discussing clearly how much of the data is base in “matched” information and the data without imputation or matching should also be available.

Match between runs (MBR) is a widely accepted method to increase the number of quantified data points in proteomic experiments, and also used in many high ranked interaction studies (Cox et al., MCP 2014, 10.1074/mcp.M113.031591, Hein et al., Cell 2015, doi: 10.1016/j.cell.2015.09.053, Eberl et al., Mol Cell 2013, 10.1016/j.molcel.2012.10.026). As we agree with the reviewer that MBR has a little but certain false discovery rate, we did not use it for the OOPS dataset, to achieve the highest stringency possible in the Student t-test comparison. For RNA-IC, however, we deliberately used MBR, as the target identification was partly based on exclusive identification in at least two out of three replicates of the crosslink pulldown, with no identification in the control similar to what has been described earlier (Sysoef et al., Nat Comm 2016, doi: 10.1038/ncomms12128). As many of the proteins in the crosslink condition are identified close to detection limit of the machine, MBR gives additional confidence verifying that the presence and absence of proteins in each group is not due to different probabilities in sequencing in a TopN method, but to the presence or absence of MS1 precursors. To allow the reader a better verification of those cases, we now added a column with the header "Identified by Match-between-runs" to the registers of "human RNA-IC Th0" and mouse RNA-IC-Th0 of Supplementary Table 1. The table therefore provides the information, which proteins were identified due to MBR, and which were found without MBR and imputation. The authors like to note, that despite many similarities in the data acquisition method, OOPS and RNA-IC are two separate methods to define the RBPome, with individual statistics and settings.

5) LFQ is usually normalised across conditions/replicates by loading equal amounts per run. How were the CL and nCL conditions normalised for fair comparison since nCL inherently recovers much less amount and diversity of

proteins? This is a crucial point to establish reliability of the RBPome catalogue and, thus, must be clearly explained and demonstrated in the main text.

Contrary to other LFQ algorithms like PSM counts, or mean intensity of TopN proteins, MaxLFQ applies pairwise comparisons of peptides, and protein intensities are calculated from the median of all possible comparisons. As in affinity purification experiments, the interactors are mostly unidentified in the control condition, run to run normalization is predominantly based on the population of background binding proteins, which are present in control and bait conditions. Therefore, MaxLFQ can efficiently cope also with difficult to normalize scenarios such as OOPS and RNA-IC.

Even if one would assume that a normalization bias would be generated by different amounts of proteins in CL and nCL conditions, this would actually lead to an underestimation of interactors, given that the LFQ values proteins in the control group would increase due to the normalization, and therefore the fold enrichment decrease. Therefore, we consider the risk of introducing false positive interactors by normalization bias to be considerably low.

We included an explanation of the normalization by MaxLFQ together with a reference of the paper describing the algorithm in the method section: "Protein intensities were calculated using the MaxLFQ algorithm⁶⁷, which is based on pairwise calculation of peptide ratios."

6) The authors do not make clear why they used two separate RNA binding protein enrichment methods. OOPS gave more data than RNA-IC. If the authors considered that RNA-IC might give deeper coverage of the mRNA binding proteome than OOPS, then this should be clearly articulated. As RNA-IC does not exclusively enrich polyadenylated RNA, some clear explanation should be given. The reader at the moment can only assume that the two methods were used sequentially and OOPS possibly being an add on as a newer method.

The objective of our study was to define the RBPome of T cells to enable future analyses of specific RBP contributions and the uncovering of potential cooperations between RBPs in networks that control T cell biology. We did not intend a comparison of the two methods, since this has been done before. We are convinced that every method of identification has individual strength, bias and also limitation. We therefore felt the necessity to use more than one method, if we wanted to come up with a valid definition. Extending our initial datasets obtained by the well-accepted RNA-IC method, we chose to also utilize the most recently developed OOPS method with its particular strength being the feature that RNase-digest specifically releases RNA-bound proteins. Since these two methods utilize very different principles to enrich proteins crosslinked to RNA we thereby reach depth and breadth of RBP identification. In a similar reasoning, we not only determined the RBPome of mouse but also of human T cells. Therefore, the strength of this piece of research can be seen in the reconfirmation of RBPs through identification in different species and different methods. This makes our core RBPome trustworthy, and we think this study will be a good resource for the community of immunologists.

We have now clarified our choice of methodology in the result section by adding: "We then attempted to extend and confirm the T cell RBPome with a second approach that utilizes a different biochemical principle to experimentally enrich proteins crosslinked to RNA...."

7) Line 175-177 states that difference in the RNA-IC derived RBPome for different types of T cells could have arisen from an incomplete assessment of the RNA-IC methods. I don't follow this argument as the same method was used for all - it would suggest the replicates are variable rather than incomplete assessment.

We agree and have changed our interpretation: "Accordingly, these findings suggest that iTreg cells as an example for T helper cell subset specialization differentially express eight mouse and 20 human RBPs that do not overlap between the two species (Supplementary Fig. 2f)."

8) I would like to have seen a fuller assessment of the types of proteins enriched by RNA-IC method but not OOPS. Given OOPS is more sensitive overall, perhaps proteins enriched by RNA-IC over OOPS have specific physical properties and/or biological roles.

For both mouse and human datasets, between 60-70 RBPs were uniquely identified by RNA-IC. We did not find any specific physical properties for this group of proteins, as for example no significant difference in content of intrinsically disordered regions.

We also did not find any biological roles specific for these proteins. In fact, RNA-related biological roles are generally less enriched compared to proteins identified by both RNA-IC and OOPS.

These results are in agreement with a previous comparison of RNA-IC and Trizol extracted RNA-crosslinked proteins (Trendel et al., Cell 2019), which also did not identify specific functions for the RNA-IC only group. For clarity reasons, we decided to not include the comparison in our study.

9) In the methods section, the authors cross-link Tefr or iTreg cells at 0.2 J/cm² at 254nm for 1 min in the RNA-IC section. In the OOPS section, they claim that the cells 'floating on ice' were irradiated once at 0.4 J/cm² and twice at 0.4 J/cm² at 254nm (for who long?). There is no mention of why these two different sets of parameters were used, why the CD4+ T cells required additional cross-linking or how these parameters were optimized. This is crucial, as over irradiation may lead to additional stress responses in the system,...

The irradiation time (~1 min 15s for 200 mJ/cm and accordingly ~ 5 min for 2x200 and 1x400 mJ/cm²) should not have been mentioned, since we programmed the crosslinking device to reach the specific energy per cm² and not energy per cm² per time. We have changed this inconsistency now.

... and insufficient cross-linking will lead to inefficient capturing of the RBPome. Moreover, in the OOPS methods section the authors carry out five successive rounds on phase partitioning, with no mention of why they chose this number and what benefit additional rounds over and above what reported in the original OOPS paper (Queiroz et Nat. Biotech 2019) was observed. Given that UV crosslinking needs to be optimized for each cell line and there is limited data for UV crosslinking efficiencies for T cell RBPs, I was surprised that no optimization data was proffered in this manuscript.

For RNA-IC we tested doses around 150 mJ/cm², which is typically used for adherent cell lines. We determined that the applied dose of 200 mJ/cm² UV light for suspension T cells allows the capture of proteins of all sizes (as determined by silver stain), the capture of specific RNA-binding proteins and the capture of specific mRNAs but not rRNAs (shown in Supplementary Fig. 1c-e).

When establishing the OOPS method in our lab we followed the suggested experimental determination of UV-induced removal of RNA from the aqueous phase (Queiroz et al., Nat. Biotech., 2019). To do so, the UV dose was

selected that removed approximately 75% of the non-crosslinked RNA from the aqueous phase. The respective optimization experiment is now shown as Supplementary Fig. 3a.

“Evaluating the method, we selected the UV dose that removes 75% of the total RNA in the aqueous phase ²⁹ (Supplementary Fig. 3a) and investigated selected RNA and proteins from purified interphases derived from CL and nCL MEF cell samples (Supplementary Fig. 3b-c).”

Minor points

Line 37: ‘including Roquin’ – you should be clear whether you are talking about Roquin 1 or 2 or both

We have shown that both proteins work redundantly in T cells (Vogel et al., 2013). In addition, their binding to RNA depends on an almost identical polypeptide sequence of the ROQ domain (Schlundt et al., 2014). We intended to facilitate the descriptions by talking about “Roquin”. However, we agree that this is not precise and future experiments may reveal differences about the two paralogs. We have therefore now changed the previous nomenclature of ‘Roquin’ to ‘Roquin-1 and Roquin-2 or Roquin-1/2’ throughout the manuscript.

Lines 49-53 would benefit with some additional references

We included the requested references on T cell biology (See Ref 1-7).

Line 55: Previous studies in T helper cell

We changed the sentence accordingly.

Line 64: Which Roquin binds together with Nufip2 to RNA?

Both Roquin proteins bind to Nufip2, but we only tested binding in a ternary complex with RNA using a fragment of Roquin-1 (aa2-441). Therefore, most likely both Roquin proteins bind with Nufip2 to RNA, but formally this was only proven for Roquin-1 (Rehage et al., 2018). We adjusted the sentence accordingly.

Line 75. It is false to claim RNA-IC identifies “exclusively” poly-A RNAs since it also identifies ribosomal proteins for instance as clearly observed in published studies.

We admit that our wording was not precise. While the RNA-IC method certainly was designed to identify mRNA interactomes (Castello et al., Cell, 2012) it has been shown that this exclusivity is not the case (references were also kindly provided by reviewer 3). We reworded the sentence: ...however, it is constricted by design, intending to identify proteins binding to polyadenylated RNAs.

Line 77: Where is the original OOPS citation?

We agree that this reference should also be cited here.

Line 105: Which microRNAs repress Roquin 1 and 2. Define a TTP binding site?

We changed the original sentence to avoid such misunderstanding: The ICOS mRNA has a long 3'-UTR, which responds in a redundant manner to Roquin-1 and Roquin-2 proteins. ICOS expression is also repressed by Regnase-1 ^{15,18} and by microRNAs ^{41,42}. In addition, sites of TTP binding in the ICOS mRNA have been determined by crosslinking and immunoprecipitation ¹². Moreover,...

Line 143: “To decipher the post-transcriptional the network” . remove “the” before “network”

We changed the sentence accordingly.

Sections “The polyadenylated RNA-bound proteome of mouse and human T helper cells” and “the global RNA-bound proteome of mouse and human T helper cells” : the number of cells and their state should be reported in the main text

The number of cells has now been added to the main text and a phenotypical description of their state is shown in Supplementary Fig. 2a-d.

Line 180: Insert OOPS citation

We inserted the citation.

Line 211: ‘In comparison to single shot total proteomes’...not clear what is mean by single shot total proteomes – no replicates? single injection on LC-MS/MS??

We agree with the reviewer that the wording was not precise and rephrased it to: “In comparison to total proteome measurements.” Single-shot proteomes are to be seen in contrast to fractionated proteome measurements, which were not present in this study. Whole proteome measurements were performed on a small fraction of the T cells that were used for RNA-IC and OOPS experiments and hence have the same numbers of replicates.

Line 224: what is meant by (comprises 26 to 224 members)? I assume the authors mean RNA binding domains, but does the number refer to different domains or the total number of proteins? For example, a protein might have two KH domains, so is this counted as 1 or 2 members?

These results refer to the annotations of the EuRBPDB (J. Y. Liao, NAR, 2020) and the type of RBD defines a member of an RBP family. This means that one protein with two different types of RBDs is part of two families, while the repeated presence of one type of RBD does not lead to multiple listings.

Preparing the answer to this reviewers minor point we noticed that our results were based on EuRBPDB tables of annotated RBPs that included only one RBD for each RBP. Realizing this shortcoming we repeated our analysis. This time the results are based on comparisons between RNA-IC/OOPS-identified RBPomes and RBP family members. While this new analysis takes into account that one protein can contain more than one RBD and hence be part of more than one RBP family, the overall results hardly changed. The number of identified RBDs was identical for the OOPS-identified RBPomes and slightly higher for RNA-IC-identified RBPomes. We changed Fig. 4d and Fig. 4f to represent the more accurate results. This new analysis also had a slight impact on Supplementary Fig. 5, which we also updated.

Line 227: ‘These findings underscored that the OOPS method recovered RBPs from additional, non-polyadenylated RNAs and that RNA-IC-derived RBPomes are specific but incomplete’. This is inaccurate...the OOPS method is more efficient and may therefore also capture additional mRNA interacting proteins and RNA-IC derived RBPomes are not completely specific as reported in many previous RIC publications.

We agree that RNA-IC is not completely specific and that the OOPS method is more sensitive. We maintain however that many CD4⁺ T cell RBPs were identified by OOPS as well as by RNA-IC. We have changed the sentence to now hopefully include the relevant aspects: These findings underscored that the OOPS method is likely more sensitive and by design recovered RBPs from additional, non-polyadenylated RNAs and that our RNA-IC-derived RBPomes are mostly specific but incomplete.

Line 235: what do the authors mean by false positives? Proteins that partition to the interphase during the OOPS

protocol and are released by RNase treatment despite not binding RNA? Proteins that bind indirectly to RNA binding proteins during both enrichment protocols used?

We agree that false positive results would be hard to explain and have removed this interpretation that was included out of caution.

Line 285: if Roquin -1 binds RNA in one part of the cell and not in elsewhere how do you know are that your interaction list is only involved with the Roquin 1 RBP interaction complex?

We don't see the list of proteins derived from BioID as physical but rather functional interactors, as we find these proteins in close proximity of an RNA-binding protein and the majority of these proteins contain RBDs themselves. We think that many of these proteins bind independently on adjacent composite cis-elements, and therefore designed the small-scale screen for cooperative or antagonistic activities in concert with Roquin-1.

Line 293: What is meant by 'Reflecting deletion'? This is poor scientific language

We deleted "reflecting deletion".

Line 322: Atlas would suggest some spatial information - as total cell lysates were used here, what the authors have created is a list not an atlas.

We deleted "atlas".

Line 333: what do they mean by "false-positive proteins"? do the authors believe that the controls used are not adequate?

We admit that this was too speculative and have deleted this part of the sentence.

Line337: define and justify 'cut-off'

The cut-off was defined as 0.05 FDR, >2fold enrichment, we have indicated this now.

Line 354-355: This sentence does not make sense.

We admit that this was too speculative and deleted the sentence entirely.

Line 415: 'squished' is not an especially precise scientific term!

We changed it to single cell suspensions were generated from lymphoid organs and passed through....

Line 518-9: What do the authors mean when they say 'activated without bias'?

We mean and have now corrected the statement that CD4⁺ T cell were activated without cytokines and antibodies that skew their differentiation into specific T helper cell subsets to generate unskewed Teff cells.

Line 621: 'width of 0.2 (0.25 for OOPS data),' What does this mean and how can the authors use different parameters for the two different datasets – where is the statistical justification?

This refers to the width of the normal distribution from which random numbers are taken to replace missing values.

The width is 0.2x (0.25x) the standard deviation of the actually measured data.

This value varies between 0.2 (Hemmer et al., Nat Comm 2019, doi: 10.1038/s41467-018-08196-5) and 0.3 (Hubner et al., JCB 2010, doi: 10.1083/jcb.200911091) in different studies. As indicated before, OOPS and RNA-IC are independent datasets, which have different ways to define RNA binding proteins.

To verify that this value has not a big difference on the number of interactors, we reanalyzed RNA-IC with width value of 0.2 and 0.25. Using a width of 0.2 for RNA-IC data identified 4 (mouse) or 3 (human) more proteins, which are all previously known RNA-binding proteins.

Reviewer #2 (Remarks to the Author):

Hoefig et al. present a comprehensive study shows that Icos responds from several RBPs, including Roquin, Regnase-1, Wtap, Dgcr8. They found that hundreds RBPome of mouse and human T cell proteins by RNA-IC and OOPS. Interestingly, they also found STAT1, STAT4 as the core proteome unexpectedly. Also, they identified RBPs which are related with Roquin-1 in CD4+ T cells. The paper is clearly written and technically sound. There are no

ethical concerns arising. The methods are properly conducted, statistical analysis of data is sound and claims are appropriately discussed. The authors' claims are mostly convincing.

We thank this reviewer for expressing enthusiasm for our study. We have included experimental evidence and explanations to address the minor points.

I have only some minor suggestions below.

1. I did not see any tables (table 1, supplementary table 1, 2, 3). Please make sure that.

We are deeply sorry for neglecting to include the supplementary tables in the first submission. However, it was an honest mistake, as evident from us making these Excel files freely available on bioarchives (<https://www.biorxiv.org/content/10.1101/2020.08.20.259234v1.supplementary-material>) on August 20th, 2020.

2. In figure 1c, f, i, l, I can see decreased each protein level, but I cannot judge deletion efficiency. It is better to show with immunoblot.

*We are now also providing immunoblots according to this suggestion, which demonstrate near complete deletion of the inducible knockouts (new **Supplementary Fig. 1a**). Please note that the minor shift observed for Regnase-1 deletion (**Fig. 1f**) is explained by low basal expression of this inducible gene product in resting T cells and does not indicate inefficient deletion by the tamoxifen induced Cre recombinase.*

3. In figure 1, Did you see any changes phenotype of T cells by deletion of Roquin, Regnase, Wtap, or Dgcr8?

While preparing these experiments, we have tried to see and documented such effects. The figure below shows one exemplary anti-CD44 staining of one set of mice used in Fig. 1a-g of the manuscript. We find comparable CD44 activation marker expression of T cells lacking Roquin-1/2, Regnase-1 and Wtap inducible deletion compared to WT T cells. The genetic deletion of these genes by non-inducible Cre recombinases obviously imposes a number of phenotypes already during T cell development, homeostasis and challenge, as documented by many publications from our as well as other labs. However, inducible deletion by CD4-Cre-ERT2 for the shortest possible period of time, as we employ here, does not alter the cells in a profound way, and the majority of T cells is still naive (i.e. CD44^{neg}, see below black/grey curves) on d0 and can be similarly activated, thereby becoming CD44^{hi} during in vitro stimulation with plate-bound antibodies on d1 and d2 for all the tested genotypes (see red and blue curves on d1 and d2 respectively). Please also see Zeiträg et al., Eur. J. Immunol 2021 for the effect and phenotype of T cells with similar CD4-Cre-ERT2-induced deletion of Dgcr8.

Inducible deletion of Roquin-1 and Roquin-2 or Regnase-1 or Wtap encoding genes does not impair in vitro activation as determined by CD44 expression. The grey curve indicates the phenotype of CD4⁺ T cells isolated from mice, which received tamoxifen gavage to induce the activity of Cre-ERT2 expressed from the CD4 locus,

while the red and blue curves reveal anti-CD3 and anti-CD28 dependent induction of CD44 on day 1 (red) and day 2 (blue) as a measure of T cell activation.

4. Why did you use Th1 subset for this study, not others? If you use other subset, do you think you will see same result?

In this experiment we utilized Th1 instead of unskewed effector T cells, because C57BL/6 mice have a bias towards differentiation into Th1 cells. Therefore, most knockout T cells can be skewed equally well into this subset (at optimal IL-12 concentrations and with strong stimulation by plate-bound anti-CD3/anti-CD28 antibodies). If we had used the Th2 cells or Th17 subsets, the deficiency of Roquin-1/2 encoding genes, would have counteracted or enhanced the differentiation itself (Jeltsch et al., 2014), which in turn is known to affect ICOS levels. Nevertheless, we know that the strong direct regulation of ICOS by Roquin can be demonstrated in all T helper cell subset inducing conditions and also CD4⁺ or CD8⁺ lineages (Glasmacher et al., 2010, Jeltsch et al., 2014 and unpublished information).

5. What is definition of Teff cells? Please clarify.

We used this term to indicate that we allowed T cells to become activated and adopt an effector memory phenotype without skewing them experimentally towards a specific subset. However, for our genetic background of mice and also for human T cells, the major subset specific cytokines that these cells express upon restimulation is IFN- γ . We have now added this or similar explanations: "... CD4⁺ T cells were isolated as described above and activated without cytokines and antibodies that skew their differentiation into specific T helper cell subsets to generate unskewed Teff cells." in the main text and methods section.

6. In line 427, "~ initial cell density of 5 or 1.5x10⁶ cells/ml" why did authors culture in 5x10⁶ cells/ml? It is not usual. In that high concentration, cells cannot be activated properly. Does it mean 0.5? Please explain that.

We have started to use this high concentration only when performing retroviral transductions (saving antibodies and cytokines and leaving enough volume for retroviral supernatants, without imposing negative effects on viability or activation). This treatment became possible due to the exact timing of the next step to exactly 40h of activation, as the cells would exhaust the media upon cell division. At this time point under our cell culture conditions the T cells are covering the entire bottom of the plate (maximizing the numbers of infected cells), are fully viable (please note that before the biggest parameter of cell death was the removal of media and replacement with retrovirus containing supernatant) and are met by the retroviruses just before their first cell division (which is a requirement for retroviral integration). Under these conditions and owing to the precise timing we can obtain retroviral transduction efficiencies of more than 95% (with empty retroviral vectors). We have improved our description: seeded T cells on goat α -hamster IgG (MP Biochemicals) pre-coated 6-well (5 Mio cells/mL for 40 h for transductions with retroviruses) or 12-well (1.5 Mio cells/mL for 48 h for expression analyses) plates.

Reviewer #3 (Remarks to the Author):

In this study, the authors identified the core RBPome in mouse and human in T cells by performing both oligo-dT capture and orthogonal organic phase separation (OOPS). The authors found STAT1, STAT4, and Vav1, proteins that had been characterized as transcription factors (TFs) or a guanine nucleotide exchange factor, in the core RBPome, and the tethering assay revealed that these proteins are potent to downregulate gene expression by directly binding with RNA. Then the authors tried to identify Roquin binding proteins comprehensively by performing BioID experiments, which resulted in the identification of about 50 proteins as RBPs in proximity to Roquin-1. Among them, the authors found that a set of RBPs, including Rbms1, Cpeb4, Igf2bp3, and Celf1, could modulate the

expression of a set of Roquin-1 target mRNAs. Collectively, the authors conclude that the different proteomic approaches could determine higher-order functional RNA-protein interactions.

Overall, this manuscript is largely descriptive and preliminary due to the lack of mechanistic analysis for the functions of newly identified RBPs. This manuscript is comprised of two parts, the identification of RBPome in T cells and the identification of Roquin-associating proteins, though the relationship between these two stories is not clear. It is rather recommended to separate them into two different manuscripts.

Specific comments are shown as follows:

We thank this reviewer for helping to improve the manuscript in its current form.

Major comments

1. In Figure 1, the authors argue that there is a coordinated higher-order regulation by several RBPs, focusing on Icos expression. Although Roquin and Regnase-1 had been shown to directly regulate Icos expression, it is not clear if the lack of Wtap or Dgcr8 directly controls Icos mRNA via m6A methylation or microRNAs. Alternatively, the impaired activation of T cells due to the lack of the genes resulted in the altered expression of Icos. Furthermore, there is no evidence that RBPs listed in Fig. 1m-q directly control Icos expression. Thus, the data are not sufficient to claim that there are higher-order post-transcriptional regulation just by monitoring the expression change in a set of RBPs and Icos upon T cell activation.

We agree with the assessment of this reviewer that Fig.1 cannot make the claim of proving higher order regulation of Icos by some or all of the genes or proteins analyzed. We showed this as an example for complex post-transcriptional regulation and describe the impact of deletion of different RBPs on ICOS expression as well as their respective expression in a temporally resolved manner.

In contrast to this reviewer, we do not think that the observed effects can be explained by inefficient activation of the knockout T cells, since this would rather impact on the steep induction of ICOS (100-fold from d0 to d1 and d2), which is not impaired in any of our knockout lines (see Fig. 1a, d, g, j). In addition, while preparing these experiments, we have documented and show exemplarily (below) anti-CD44 staining of one set of mice used in Fig. 1a-g of the manuscript. We find comparable CD44 activation marker expression of T cells lacking Roquin-1/2, Regnase-1 and Wtap inducible deletion compared to WT T cells. The genetic deletion of these genes by non-inducible Cre recombinases obviously imposes a number of phenotypes already during T cell development, homeostasis and challenge, as documented by many publications from our as well as other labs. However, inducible deletion by CD4-Cre-ERT2 for the shortest possible period of time, as we employ here, does not alter the cells in a profound way, and the majority of T cells is still naive (i.e. CD44^{neg}, see below black/grey curves) on d0 and can be similarly activated, thereby becoming CD44^{hi} during in vitro stimulation with plate-bound antibodies on d1 and d2 for all the tested genotypes (see red and blue curves on d1 and d2 respectively). Please also see Zeitrög et al., Eur. J. Immunol 2021 for the effect and phenotype of T cells with similar CD4-Cre-ERT2-induced deletion of Dgcr8.

Inducible deletion of *Roquin-1* and *Roquin-2* or *Regnase-1* or *Wtap* encoding genes does not impair *in vitro* activation as determined by CD44 expression. The grey curve indicates the phenotype of CD4⁺ T cells isolated from mice, which received tamoxifen gavage to induce the activity of Cre-ERT2 expressed from the CD4 locus, while the red and blue curves reveal anti-CD3 and anti-CD28 dependent induction of CD44 on day 1 (red) and day 2 (blue) as a measure of T cell activation.

In **Fig.1** of the manuscript we want to raise the awareness for the potential of higher order regulation of targets like ICOS by different RBPs (*Roquin-1/2*, *Regnase-1*, m6A-binders and miRNAs) and through yet unknown factors of the RBPome, which could also dynamically change according to the regulation of expression and post-translational modification of the trans-acting factors.

To prevent a misunderstanding, we changed the wording of the last sentence describing Fig. 1: "Together these data indicate that mRNA targets can respond to simultaneous inputs from several RBPs, which are aligned by dynamic expression and post-translational regulation and can orchestrate redundant, cooperative and antagonistic effects into a coordinated higher order regulation."

Nevertheless, examples of higher order regulation of ICOS have been reported including the regulation by *Roquin-1* and *Ago2/miR-146a* (Srivastava et al., 2015) as well as by *Roquin-1* and *Nufip2* (Rehage et al., 2018) and may also include the observed cooperation of *Roquin-1* and *Regnase-1* (Jeltsch et al., 2014, Jeltsch and Heissmeyer 2016), and, as suggested by our new data, by *Roquin-1* and *Celf1*. We can now demonstrate that *Celf1* and *Igf2bp3* colocalize with *Roquin-1* in P bodies and demonstrate *Celf1* binding to the *Icos* and *Ctla4* but not to the *Nfkbid* mRNA in T cells, which nicely correlated with the observed effect of *Celf1* overexpression on *Roquin-1* mediated *Icos* and *Ctla4* but not *Nfkbid* repression (see new **Fig. 7e-f**). Of note, the future mapping of binding sites for these RBPs on the *Icos* mRNA in T cells as well as sophisticated structural and genetic approaches will be required to demonstrate the mechanistic basis.

2. The authors determined RBP profiles in mouse and human T cells, which resulted in the identification of 300 and 1000 RBPs by oligo-dT and OOPS methods, respectively (Figures 2-4). Although this is a major part of this manuscript, the authors just analyzed GO enrichment or protein recovery. It is interesting to compare the RBPome of different cell types to analyze if the RBPome contains cell type-specific RBPs.

We agree and have performed such an analysis, have included a new Fig. 4h and discussed the findings: We further compared published human OOPS data sets from embryonic kidney (HEK293), osteosarcoma (U2OS) and mammary epithelial (MCF10a) cell lines²⁹ with our dataset from primary human CD4⁺ T cells (**Fig. 4h**). The 4-way comparison shows that although similar numbers of RBPs were identified overall, the number of uniquely identified RBPs was almost three times higher in CD4⁺ T cells than in each of the cell lines (**Fig. 4h**, left panel). Of the 439 CD4⁺ T cell-unique RBPs 294 were newly discovered and 145 were previously annotated (**Fig. 4h**, right panel and **Supplementary Table 1**). The annotated RBPs can be further divided into 92 canonical and 53 non-canonical RBPs (**Supplementary Table 1**). We interpret the result such that RBPomes are strongly affected by tissue-specific

expression of RBPs and/or RBP activity in the presence or absence of post-transcriptional modifications, substrates and co-factors.

3. In Figure 5, based on their RBPome data, the authors identified Stat1 and Stat4 as potential RNA-binding proteins. Since UV crosslink at 254 nm can irreversibly bridge the interaction between proteins and nucleic acids, DNA is possibly contaminated in this assay, resulting in the identification of DNA-binding proteins (Conrad T., et al., Nature Communications 2016, Perez-Perri J., et al., Nature Communications 2018). To exclude this possibility, the authors need to show more direct evidence as to the interaction of these Stat proteins with mRNA such as the EMSA assay by using recombinant proteins.

We took the advice and purified several mouse and human Stat (Stat1, Stat3, Stat4) proteins, and also included a step that removes associated DNA and RNA from E. coli. According to the previous work that we referred to in our discussion section, Stat proteins may be able to interact in a sequence specific manner with the TSU ncRNA, at least when performing RNA-EMSA with extracts from cell lines transfected with the Stat1 protein (Peyman JA, Am J Reprod Immunol., 2001). To verify and extend these findings, we tested direct physical interaction of in vitro transcribed TSU with purified Stat proteins. As we demonstrate in RNA-EMSA for Stat1 and Stat4, there was a specific interaction with the TSU lncRNA also for purified proteins, which resisted competition through an excess of yeast tRNA (new Fig. 5c). We included the statement:

"We then tested human and mouse STAT1 and STAT4 proteins as purified recombinant proteins in RNA-EMSA with in vitro transcribed TSU lncRNA. This lncRNA is expressed in human cells and early work involved a sequence-specific recognition through Stat1-transfected cell extracts^{54,55}. Performing RNA-EMSA without and with competitor RNA we determined binary interaction of mouse and human STAT1 and STAT4 that was at least partially resistant to unspecific competition (Fig 5c). Our results thereby excluded a requirement for additional factors or signal-induced Stat protein modification in eucaryotic cells or any indirect contribution from cell extracts in these RBP/RNA interactions. These findings support a potential moonlighting function of these signaling proteins. "...

We therefore think to have proven the binding of STAT1 and STAT4 to RNA. In general, RBP identification by the OOPS method does not only require the protein to be crosslinked to RNA, or potentially off-target to DNA. It also needs to be specifically released from the RNA-protein adduct upon RNase treatment. This step should in all likelihood prevent the identification of proteins that were crosslinked to DNA.

4. The authors tethered Stat proteins to the 3' UTR of luciferase mRNA by using the λ N-BoxB system, and claim that Stat1 and Stat4 harbor RNA regulatory functions. However, the system is artificial and preliminary. First, it is not clear if Stat1 and Stat4 regulate endogenous mRNAs. Second, the authors need to clarify how Stat1 and Stat4 suppress protein production. Do they induce degradation of mRNA or inhibit protein translation?

To address this point, we repeated the tethering experiment which now became the new Fig. 5g and analyzed the transcript levels of the luciferase mRNAs by quantitative PCR. We can now demonstrate that the observed effect on the renilla luciferase activity/ protein was equally present on the renilla mRNA levels, which suggested the regulation of mRNA abundance due to Stat protein tethering. Although we do not know the physiologic target(s) of Stat1 or Stat4, we can demonstrate that their repressive effect was comparable to that of the prototypic positive control Pat1b.

We changed the description of the experiments accordingly: "Importantly, Stat1 and Stat4 repressed luciferase function almost to the same extent as the known negative regulators Pat1b and Roquin-1, or other known RBPs, such as Celf1, Rbms1 and Cpeb4 (Fig. 5g), and this repression reduced the abundance of the boxB containing renilla luciferase mRNA (Fig. 5h)."

5. It is also interesting and important to examine if the stimulation of T cells with IFN or IL-12 alters the Stat-mRNA binding capacity and the activity to suppress protein production.

Since we find direct interaction with RNA of Stat1 and Stat4 proteins purified from bacteria, we think that this activity does not require IFN or IL-12 induced phosphorylation of the proteins as suggested by this reviewer.

6. Do other Stat proteins also function as the RBP?

We think that the answer to this question will be yes, since we find more Stat proteins in the different RBPomes (see Table below). So far strong evidence only involves Stat1 and Stat4 in RNA binding and regulation. In addition, we cloned and tested mouse Stat3 in EMSA. Since this protein preparation was not as pure, less concentrated and also induced a less intense band-shift, we suspect Stat3 to also bind, but we cannot show a result with high confidence at this point.

Stat proteins identified in RBPomes:							
TH0							
Mouse RNA-IC	Stat1			Stat4			
Human RNA-IC	(STAT1)						
Mouse OOPS	(Stat1)	(Stat2)	Stat3	(Stat4)		Stat5b	Stat6
Human OOPS	Stat1		STAT3	Stat4	STAT5a	STAT5b	(STAT6)
	(…) = identified but not significant						

7. In Figure 6, the authors took advantage of the Bio-ID system to identify proteins in proximity to Roquin-1. However, data shown in Figures 6 and 7 are independent of the data in Figures 1-5. Thus, it is rather better to publish them as two distinct studies.

The BioID experiment was performed to enable a smaller-scale screen of RBPs that functionally interact with Roquin-1-mediated post-transcriptional gene regulation. Since primary T cells are not amenable to high-throughput screening approaches (i.e. all 1000 RBPs), we intended to focus our analysis only on RBPs in close proximity of Roquin-1. Therefore, this data set has enabled us to apply the information of the T cell RBPome on a specific biological question.

8. The BioID experiment identified only 64 and 143 Roquin-1 proximity proteins in primary CD4 T cells and fibroblasts, respectively. Surprisingly, the lists are not so overlapping between T cells and fibroblasts. Does this mean the Roquin-1 complex differs depending on the cell type? Or is this because the experiments did not reach saturation? The proteins checked by the authors in Figure 1m seems to be detected only in MEFs by the BioID assay. The authors need to explain the reason why they focused on these proteins.

Performing BioID in MEF cells allowed stable transduction of BirA-Roquin-1, selection with puromycin and generation of cell clones enabling optimal expression of the bait in all cells. In contrast, the limited lifespan of primary T cells only allows retroviral transduction for short term expression and bulk analyses. Despite these experimental differences we would argue that there is a good overlap for the proteins that have been identified with the highest confidence. However, we did not concentrate only on this "top of the list", since we noticed that also important cofactors like Edc4 and Nufip2 were identified with somewhat lower confidence. Different from this reviewer, we do not interpret the lists as interactors within one stable Roquin-1 complex, but rather as different RBPs binding in vicinity on the same mRNAs. Obviously either or both interpretations can be true. Therefore, differences in the BioID*

lists between MEF and T cells may be explained by differences in complex formation, differences in the cellular RBPomes and differences in the composite cis-elements present in the different transcriptomes of both cell types. There is a misunderstanding: We did not focus on the proteins from Fig. 1m, but tested all proteins overlapping in BioID experiments but then additionally included all MEF cells BioID hits, which were part of the core RBPome, (minus those that we were unable to clone and those that had cDNAs too large to be expressed in retroviruses or those that did not express, as judged by an absence of GFP-positive cells). To account for the complex set-up (overexpression in WT or inducible KO) and the use of limited primary CD4⁺ T cells (from a mouse line with compound (i.e. 6!) genetically modified alleles), we performed 5 subsequent experiments and tested partially overlapping subsets of these 46 candidates, each time including the hits from the preceding small-scale screen in the subsequent one to judge reproducibility.

9. The authors examined the localization of the candidate protein transfected with the corresponding GFP-fused constructs in HEK293T cells (Supplementary Fig 7c). However, they failed to show the localization of Roquin-1 in the same cells to claim the close proximity of the Roquin-1 to the protein of interest.

To address this criticism, we now demonstrate the proximity of GFP-Celf1 and GFP-Igf2bp3 with mCherry-tagged Roquin-1 in BFP-Ddx6 identified P-bodies. We did not perform this analysis for all other candidate proteins as they did not reveal Roquin-dependent regulation of targets.

10. In Figure 7, the authors chose four RBPs (i.e. Rbms1, Cpeb4, Igf3bp2, and Celf1) out of 46 candidate proteins, and tested if these RBPs can modulate Roquin-targeting mRNAs under wild-type and Roquin-deficient conditions. *We would like to correct that all 46 candidate proteins were tested. We also analyzed re-introduced Roquin-1 as a positive control and GFP only as a negative control. However, except for the four proteins the 42 remaining proteins did not exhibit significant reproducible regulation of Icos (as we state in the text on page 11), with the exception of Rbms1, Cpeb4, Celf1 and Igf2bp3.*

11. They finally concluded that there are higher-order functional interactions that is Roquin-dependent. However, the data are so preliminary and descriptive, and they are not so informative without further detailed analysis. For instance, they show that Rbms1 expression upregulated Ctla4, but not some of the other Roquin-1 target mRNAs, irrespective of the presence of Roquin proteins. However, this study lacks the mechanistic analysis of how Rbms1 specifically controls Ctla4 expression. And also, it is not clear the reason for the presence of Roquin-1 in close proximity to Rbms1. Similarly, the authors showed that the expression of Igf2bp3 and Celf1 downregulated the expression of Roquin-targeting genes in wild-type, but not in Roquin-deficient cells. However, they failed to show the relationship between Roquin and these proteins.

We strongly agree that further investigations into the mechanisms of Roquin-1-dependent Icos regulation by Celf1 or Igf2bp3 are necessary and will in all likelihood deliver exciting results. However, we respectfully disagree that all of these investigations can be part of this publication. Here, we define the CD4⁺ T cell RBPome and provide a showcase how this information can be utilized in an RBP-centered screen to uncover previously unrecognized cooperative or antagonistic functions. Proving this point, we describe the previously unknown connections of the functions of Roquin-1 with Celf1 and Igf2bp3 through our conceptually new approach. Most importantly, we now provide evidence that Celf1 and Roquin-1 proteins are both interacting with Icos and Ctla4 mRNAs, but Celf1 does not interact with other targets of Roquin-1 (like Nfkbid). We have included this new information in the results section: "In wild-type T cells Celf1 clearly upregulated Icos, Ctla-4 and Ox40 expression but not the Nfkbid mRNA encoded IkbNS protein expression and this function was obliterated in Roquin-1-deficient iDKO cells (Fig. 7c-d). While Igf2bp3 and Roquin-1 shared a strictly cytoplasmic localization and enrichment in BFP-Ddx6-labeled P-bodies

(Supplementary Fig. 9a), the majority of GFP-Celf1 was nuclear. Only a small fraction of the protein was cytoplasmic, where it similarly colocalized with Roquin-1 in Ddx6-labeled P-bodies (Fig. 7e). The antagonistic effect could not be explained by Celf1-mediated repression of Roquin-1 on the protein or mRNA level (Supplementary Fig. 9b-c). Vice versa, Roquin-1 KO also did not affect Celf1 mRNA levels (Supplementary Fig. 9d). Instead, the observed antagonistic effect likely involved simultaneous or mutually exclusive binding of Celf1 and Roquin-1 to the same mRNAs, since we determined strong interaction of Celf1 with Icos and Ctla4 mRNAs in RNA-IP experiments, but not with the Nfkbid mRNA. In conclusion, the combination of protein-centric and RNA-centric global approaches enabled us to discover higher order functional interactions as shown for the Roquin-1/2-dependent regulation of the costimulatory receptor Icos by Igf2bp3 or Celf1."

11. Supplementary Tables 1-3 are missing in manuscript files.

We are deeply sorry for neglecting to include the supplementary tables in the first submission. However, it was an honest mistake, as evident from us making these Excel files freely available on bioarchives (<https://www.biorxiv.org/content/10.1101/2020.08.20.259234v1.supplementary-material>) on August 20th, 2020.

Minor comments

1. In Figure 1(c, f, I, and l), the efficiency of gene deletion was confirmed by flow cytometry, but negative control was missing. Especially, the deletion efficiency for Regnase-1 looks very marginal. Western blot analysis may solve the problem of high background.

*We are now also providing immunoblots according to this suggestion, which demonstrate near complete deletion of the inducible knockouts (new **Supplementary Fig. 1a**). Please note, that the minor shift observed for Regnase-1 deletion (**Fig. 1f**) is explained by low basal expression of this inducible protein in resting T cells and does not indicate inefficient deletion by the tamoxifen induced Cre recombinase or by high background.*

2. Figure legends for Figure 1 should be described precisely. The authors failed to indicate how T cells were stimulated in Figure 1 (b, e, h, and k). They described that Icos expression declined after the "removal" of the TCR stimulus, which makes data confusing as well. There is no statement on this in the Figure legend, although it was described for Figure 1m-q.

We are sorry for the confusion. We improved the figure legend for Fig.1.

3. In Figure 5a and 5b, the authors performed pull-down assays using an anti-GFP antibody, followed by western and northern blot analysis by using oligo(dT) probe. If so, the northern blotting should identify mRNAs with various length, and the blots will give smear bands if they bind with RNA without high specificity. However, the Figures indicate that there is clear "band" of mRNA following capture. Does this mean the RBPs shown in Figures 5a and 5b interact with specific mRNAs?

This experiment does not include the size separation of RNA as in the northern blot, but of the protein (fused to GFP, and without using an acrylamide gradient gel). The experiment therefore mostly indicates whether a protein is bound to RNA or not, when probing the separated proteins (covalently crosslinked with cellular mRNA) with an oligo-dT-probe.

4. In page 10 (line 245), typos North-Western blot.

We used the term North-Western to indicate a combination of the separation of proteins by Western and the subsequent detection of an unknown RNA by oligo-dT probes (Northern) at the size of the presumed RNA binding protein. We now improved the description: Using Western blotting, but detection with anti-GFP antibodies or

oligo(dT) probes we could verify pull-down of GFP-tagged proteins (**Fig. 5a** left panel) and the association with mRNA (**Fig. 5a** right panel) for the RBPs Roquin-1 and Rbms1, but also for the lactate dehydrogenase (Ldha) protein, a metabolic enzyme with known ability to also bind RNA ²⁸ (**Fig. 5a**).

REVIEWERS' COMMENTS

Reviewer #1 (Remarks to the Author):

The authors have answered the vast majority of my concerns about the first version of their manuscript. The manuscript now reads much better and essential technical details have been added. There is just one comment that I am still a little troubled by:

Original concern: Line 285: if Roquin -1 binds RNA in one part of the cell and not in elsewhere how do you know are that your interaction list is only involved with the Roquin 1 RBP interaction complex?

Rebuttal: We don't see the list of proteins derived from BioID as physical but rather functional interactors, as we find these proteins in close proximity of an RNA-binding protein and the majority of these proteins contain RBDs themselves. We think that many of these proteins bind independently on adjacent composite cis-elements, and therefore designed the small-scale screen for cooperative or antagonistic activities in concert with Roquin-1.

I think the authors have missed my point. If Roquin-1 is present in two different parts of the cell, it may interact with two completely different sets of proteins – RBPs or otherwise. Proximity tagging data will give an average of what is nearby when the bait is binding RNA and when it is not. The authors cannot say categorically that the list of proteins in close proximity to Roquin-1 when it is bound to RNA. It may have a completely different function when it is not binding RNA, but the sets of functional interactions cannot be dissected apart using the data they have collected using proximity tagging to date.

Their arguments need to be softened a little.

Kathryn Lilley

Reviewer #2 (Remarks to the Author):

I have no remarks for the revised version of manuscript by Hoefig et al "Defining the RBPome of T helper cells to study higher order post-transcriptional gene regulation". The authors have properly and fully addressed all the questions raised during the first revision.

Reviewer #3 (Remarks to the Author):

The revised manuscript is substantially improved. I believe that this manuscript is now ready for publication.